# MicroRNA miR-23a cluster promotes osteocyte differentiation by regulating TGF-β signalling in osteoblasts

Huan-Chang Zeng[1], Yangjin Bae[2], Brian C. Dawson[2], Yuqing Chen[2], Terry Bertin[2], Elda Munivez[2], Philippe M. Campeau[3], Jianning Tao[4], Rui Chen[1,2] & Brendan H. Lee[1,2]

Osteocytes are the terminally differentiated cell type of the osteoblastic lineage and have important functions in skeletal homeostasis. Although the transcriptional regulation of osteoblast differentiation has been well characterized, the factors that regulate differentiation of osteocytes from mature osteoblasts are poorly understood. Here we show that miR-23a∼27a∼24-2 (miR-23a cluster) promotes osteocyte differentiation. Osteoblast-specific miR-23a cluster gain-of-function mice have low bone mass associated with decreased osteoblast but increased osteocyte numbers. By contrast, loss-of-function transgenic mice overexpressing microRNA decoys for either miR-23a or miR-27a, but not miR24-2, show decreased osteocyte numbers. Moreover, RNA-sequencing analysis shows altered transforming growth factor-β (TGF-β) signalling. Prdm16, a negative regulator of the TGF-β pathway, is directly repressed by miR-27a with concomitant alteration of sclerostin expression, and pharmacological inhibition of TGF-β rescues the phenotypes observed in the gain-of-function transgenic mice. Taken together, the miR-23a cluster regulates osteocyte differentiation by modulating the TGF-β signalling pathway through targeting of Prdm16.

---

[1] Program in Developmental Biology, Baylor College of Medicine, One Baylor Plaza, Houston, Texas 77030, USA. [2] Department of Molecular and Human Genetics, Baylor College of Medicine, One Baylor Plaza, Houston, Texas 77030, USA. [3] Department of Pediatrics, Sainte-Justine Hospital, University of Montreal, Montreal, Canada H3T 1C5. [4] Department of Pediatrics, University of South Dakota School of Medicine, Vermillion, South Dakota 57104, USA. Correspondence and requests for materials should be addressed to B.H.L. (email: blee@bcm.edu).

Osteocytes are the most abundant cells in bone, composing ~98% of all bone cells in an individual. Until the identification of mutations in the gene coding for sclerostin (SOST) as a cause of severely progressive sclerosing bone dysplasia with autosomal-recessive inheritance[1], they had long been seen as quiescent cells trapped within the mineralized bone matrix and their critical role in regulation of bone homeostasis was underestimated. Evidence indicates that osteocytes are the major mechanosensitive skeletal cell type and have critical roles in the regulation of osteoblast and osteoclast differentiation and function[2]. However, unlike osteoblast differentiation from mesenchymal stem cells, the list of molecules and transcriptional regulators necessary for osteocyte differentiation is limited.

The importance of microRNAs (miRNAs) in bone development and homeostasis was first demonstrated by conditional ablation of Dicer, an RNase III endonuclease involved in miRNA processing, in osteoblastic lineage cells, which resulted in embryonic lethality and mineralization defects[3]. Subsequent studies have focused on understanding the physiological roles of specific miRNAs in osteoclasts and osteoblasts[4,5]. Nevertheless, the role of miRNAs in osteocyte function and differentiation is unclear.

We previously identified 20 miRNAs that were upregulated, and 14 that were downregulated, during bone morphogenetic protein 2 (BMP2)-induced osteoblast differentiation of the C2C12 cell line[6]. The miRNAs within the miR-23a cluster were among those upregulated in the miRNA microarray analysis. This cluster consists of miR-23a, miR-27a and miR24-2, which are transcribed as a single primary miRNA and subsequently processed into three mature miRNAs. Multiple studies have demonstrated an important role for this cluster in angiogenesis, haematopoiesis, osteogenesis and cancer pathogenesis[7–10]. Our finding suggests that miR-23a may be involved in the regulation of osteogenesis; however, the role of this cluster in skeletal development is unclear.

Here we report that the miR-23a cluster has important roles in the regulation of osteocyte differentiation. The physiological functions of the miR-23a cluster in the osteoblast lineage are defined using both transgenic gain-of-function (GOF) and loss-of-function (LOF) mouse models. RNA-sequencing (RNA-Seq) combined with ingenuity pathway analysis (IPA) performed on calvarial bones of GOF mice reveals that the transforming growth factor (TGF)-β pathway is one of the most affected signalling pathways, accounting for the low bone mass phenotype combined with high osteocyte density. In addition, Prdm16 is predicted to be an upstream regulator and a direct target of the miR-23a cluster by Targetscan. Overall, our work uncovers the role of the miRNA miR-23a cluster in osteocyte differentiation by regulating TGF-β signalling through its direct target Prdm16.

## Results

**Identification of the miR-23a cluster in bone**. To identify miRNAs with relevant functions in bone, we performed RNA-Seq analysis on calvarial tissue from wild-type (WT) mice (Fig. 1a) and identified the miRNA-23a (miR-23a) cluster as enriched in bone. Consistent with our finding, the miR-23a cluster was induced during C2C12 osteoblastogenesis upon treatment with BMP2 (ref. 6). To characterize the in vivo function of the miR-23a cluster in bone, we generated an osteoblast-specific GOF mouse model overexpressing the cluster under the control of the Collagen type I, alpha 1 (Col1a1) 2.3 kb promoter (Col1a1-miR-23aC; Supplementary Fig. 1a). It was observed that Col1a1-miR-23aC mice were smaller than their littermates and had compromised dentin in the incisors (Supplementary Fig. 1b), as the Col1a1 promoter is also active in odontoblasts[11]. However, the weights of Col1a1-miR-23aC mice were normalized to those of WT littermates by feeding the mutant mice with a soft chow (Supplementary Fig. 1c). In the bones of Col1a1-miR-23aC mice, mature miR-23a, miR-27a and miR-24-2 were overexpressed 2.5-, 2.5- and 4.1-fold, respectively, compared to WT littermates, as determined by quantitative real-time PCR (qRT–PCR; Supplementary Fig. 1d). Micro-computed tomography (μCT) analysis of the spine revealed that Col1a1-miR-23aC mice of both sexes had a low bone mass phenotype consisting of decreased trabecular number, decreased trabecular thickness and increased trabecular separation (Fig. 1b–d). A similar phenotype was also observed in femurs collected from Col1a1-miR-23aC mice, but the changes were milder and there were no differences in cortical thickness (Supplementary Figs 2 and 3a).

To study the effects of LOF of miRNAs, decoys (that is, RNA molecules that carry multiple miRNA-binding sites) were used to inhibit miRNA function by sequestering mature miRNAs in the cytoplasm[12,13]. To examine the physiological role of each individual miRNA in the miR-23a cluster, we generated osteogenic-specific LOF transgenic mouse models expressing decoys for miR-23a, miR-27a or miR-24-2 individually under the control of the Col1a1-2.3 kb promoter (Supplementary Fig. 1a). Col1a1-miR-23a decoy (Col1a1-miR-23D) and Col1a1-miR-27a decoy (Col1a1-miR-27D) transgenic mice were fed soft chow because of dentin defects. Similar to the GOF Col1a1-miR-23aC mouse model, Col1a1-miR-23D and Col1a1-miR-27D mice showed a low bone mass phenotype (Fig. 1e and Supplementary Figs 4 and 5). By contrast, Col1a1-miR-24 decoy (Col1a1-miR-24D) mice exhibited no significant changes in bone mass compared to their WT littermates (Fig. 1e and Supplementary Figs 4 and 5), supporting the specificity of the phenotype for miR-23a and 27a mutants. No differences in cortical thickness were seen for any of these LOF genotypes (Supplementary Fig. 3b–d). Two independent transgenic mouse lines were evaluated for each genotype, and both lines showed similar bone phenotypes by μCT (Supplementary Fig. 6). Taken together, these results suggest that miR-23a and miR-27a in the miR-23a cluster both contribute to bone homeostasis.

**The miR-23a cluster affects osteocyte differentiation**. To understand the cellular basis of the low bone mass phenotype, bone histomorphometric analysis was performed on both GOF and LOF models. Col1a1-miR-23aC GOF mice showed decreased osteoblast number and surface (Fig. 2a,b). Dynamic bone formation was assessed by double calcein labelling with a 4-day interval; both mineral apposition rate and mineralizing surface were significantly decreased and correlated with decreased osteoblast number (Fig. 2c,d). However, there were no significant changes in osteoclast number and surface (Fig. 2e,f). These data suggest that the low bone mass phenotype observed in the transgenic GOF mice is mainly due to defects in bone formation, not bone resorption. Interestingly, we also observed increased osteocyte density in spine and femur cortical bones of Col1a1-miR-23aC GOF mice compared to WT littermate controls (Fig. 2g–j). Conversely, osteocyte density in the Col1a1-miR-23D and the Col1a1-miR-27D LOF mice was significantly decreased in trabecular bones from the spine but unaltered in femur cortical bones (Fig. 2g–j). No significant changes in osteoblast numbers were seen, but the mineralizing surface was significantly decreased (Supplementary Figs 7 and 8). In contrast to trabecular bone, there was no significant change in mineral apposition rate in femur cortical bones of either GOF or LOF transgenic mice (Supplementary Fig. 9). This suggests that GOF of the miR-23a cluster can regulate osteocyte differentiation in both trabecular

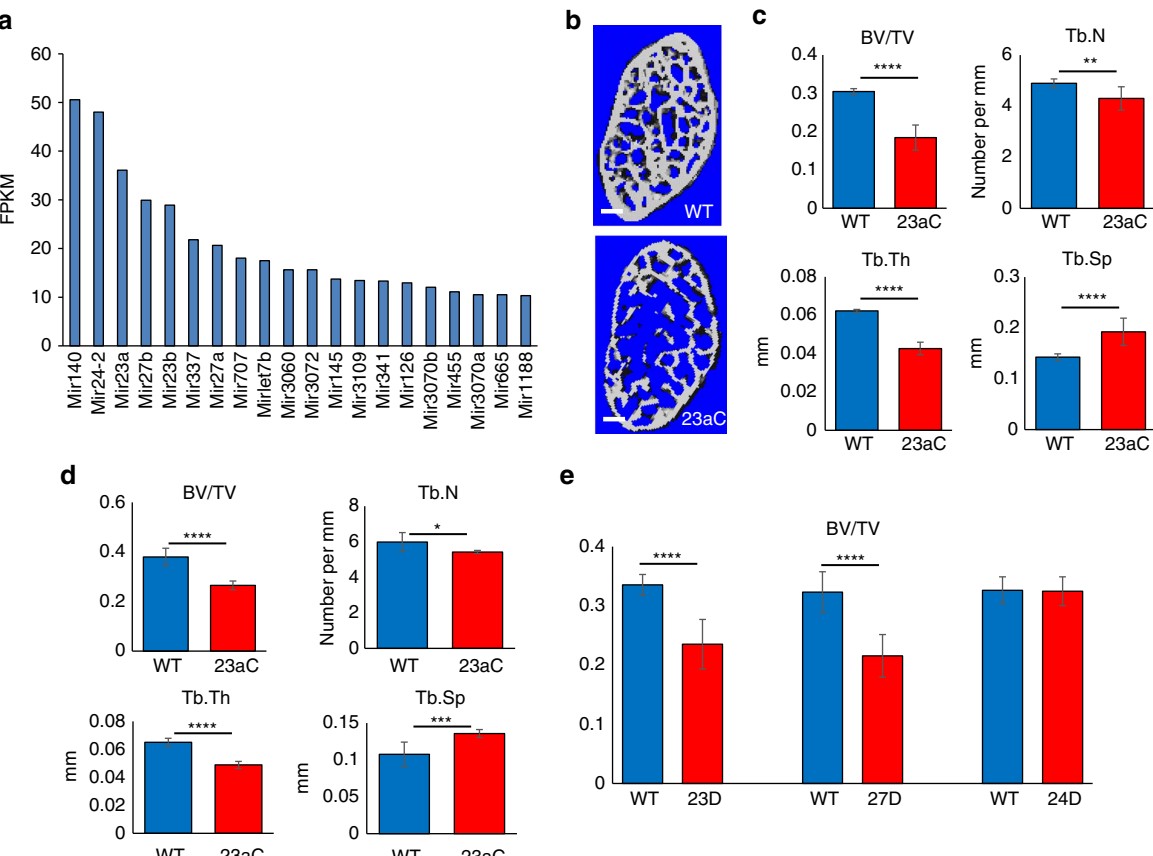

**Figure 1 | Identification of the miR-23a cluster in bone and *in vivo* function.** (**a**) Twenty most highly expressed primary miRNAs in postnatal day 3 (P3) calvarial tissue in fragments per kilobase of transcript per million (FPKM). (**b**) µCT images of lumbar spines of the osteoblast-specific GOF mouse model overexpressing the cluster under the control of the *Collagen type I, alpha 1* (*Col1a1*) 2.3 kb promoter (*Col1a1-miR-23aC*; 23aC) and WT mice (3-month-old females). White bars indicate scale of 200 µm. (**c**) µCT analysis of trabecular bone volume/total bone volume (BV/TV), trabecular bone thickness (Tb.Th), trabecular bone number (Tb.N) and trabecular bone separation (Tb.Sp) in spines of *Col1a1-miR-23aC* (23aC) and WT mice. *Col1a1-miR-23aC* mice showed a low bone mass phenotype (3-month-old female; N = 7 for WT, N = 8 for 23aC). (**d**) µCT analysis of trabecular BV/TV, Tb.Th, Tb.N and Tb.Sp of spines of 3-month-old males also showed a low bone mass phenotype in GOF mice (N = 8 for WT, N = 5 for 23aC). (**e**) µCT analysis of BV/TV in spines of LOF mouse models and WT littermates. *Col1a1-miR-23 decoy* (23D) and *Col1a1-miR-27 decoy* (27D) mice showed a low bone mass phenotype, but not *Col1a1-miR-24 decoy* (24D) mice. *Col1a1-miR-23 decoy* (N = 7 for WT, 11 for 23D); *Col1a1-miR-27 decoy* (N = 8 for WT, 11 for 27D); 24D, *Col1a1-miR-24 decoy* (N = 10 for WT, 7 for 24D). *t*-test was performed for the statistical analyses. Error bars represent the s.d. *P < 0.05, **P < 0.01, ***P < 0.005 and ****P < 0.001.

and cortical bones. In the miR-23a or miR-27a LOF models osteocyte differentiation was affected primarily in trabecular bone in the spine, but not in cortical bone in the femur. This may be due in part to the differential rate of turnover and bone formation in cortical versus trabecular bone and/or the effect size in the GOF versus LOF models.

In agreement with the reduced osteoblast activity and number observed in *Col1a1-miR-23aC* GOF mice, expression of *Satb2*, a previously reported target of both miR-23a and miR-27a identified in murine osteoblast studies *in vitro*[5], was significantly decreased in these mutants (Fig. 2k). While the levels of other early osteoblast markers (i.e., *Runx2*, *Alp*, *Osx*, *Bsp*) remained comparable to those in WT mice, we found that *Osteocalcin* (*Ocn*), a marker of mature osteoblasts, was also significantly decreased in *Col1a1-miR-23aC* mice (Fig. 2k). Consistent with the increase in osteocyte density, osteocyte markers such as *Dmp1*, *Mepe* and *Fgf23* were significantly elevated in this transgenic model (Fig. 2k). Together, these data support an accelerated differentiation of mature osteoblasts into osteocytes in the models of GOF of the miR-23 cluster.

To assess the cell autonomous effects of osteoblast-specific *miR-23aC* overexpression, we isolated bone marrow stromal cells (BMSCs) from *Col1a1-miR-23aC* and WT mice and performed

*in vitro* osteoblast differentiation. Consistent with the low bone mass phenotype observed in *Col1a1-miR-23aC* mice, Alizarin Red S staining showed impaired mineralization in the transgenic BMSC cultures (Fig. 2l,m). *Ocn* transcript levels were also decreased (Supplementary Fig. 10a). In agreement with the elevated expression of osteocyte markers observed *in vivo* (Fig. 2k), Sclerostin (*Sost*) expression was drastically increased during *in vitro* osteoblast differentiation of *Col1a1-miR-23aC* BMSCs compared to WT controls (Fig. 2n). Conversely, *Sost* expression was decreased in differentiating cultures of BMSCs isolated from the *Col1a1-miR-23D* and the *Col1a1-miR-27D* LOF mice (Fig. 2n). This increase in *Sost* with increased osteocyte density was also seen *in vivo* in femurs of *Col1a1-miR-23aC* GOF mice, while there was no significant difference in femur cortical bones of *Col1a1-miR-23D* and *Col1a1-miR-27D* LOF mice (Supplementary Fig. 11), likely due to the weaker phenotype in the LOF models. Taken together, these results suggest that the *miR-23a* cluster can accelerate the differentiation of mature osteoblasts into osteocytes.

**Prdm16 is a direct target of miR-27a and suppresses *Sost*.** To investigate the molecular mechanisms through which miR-23a regulates bone homeostasis and terminal differentiation of

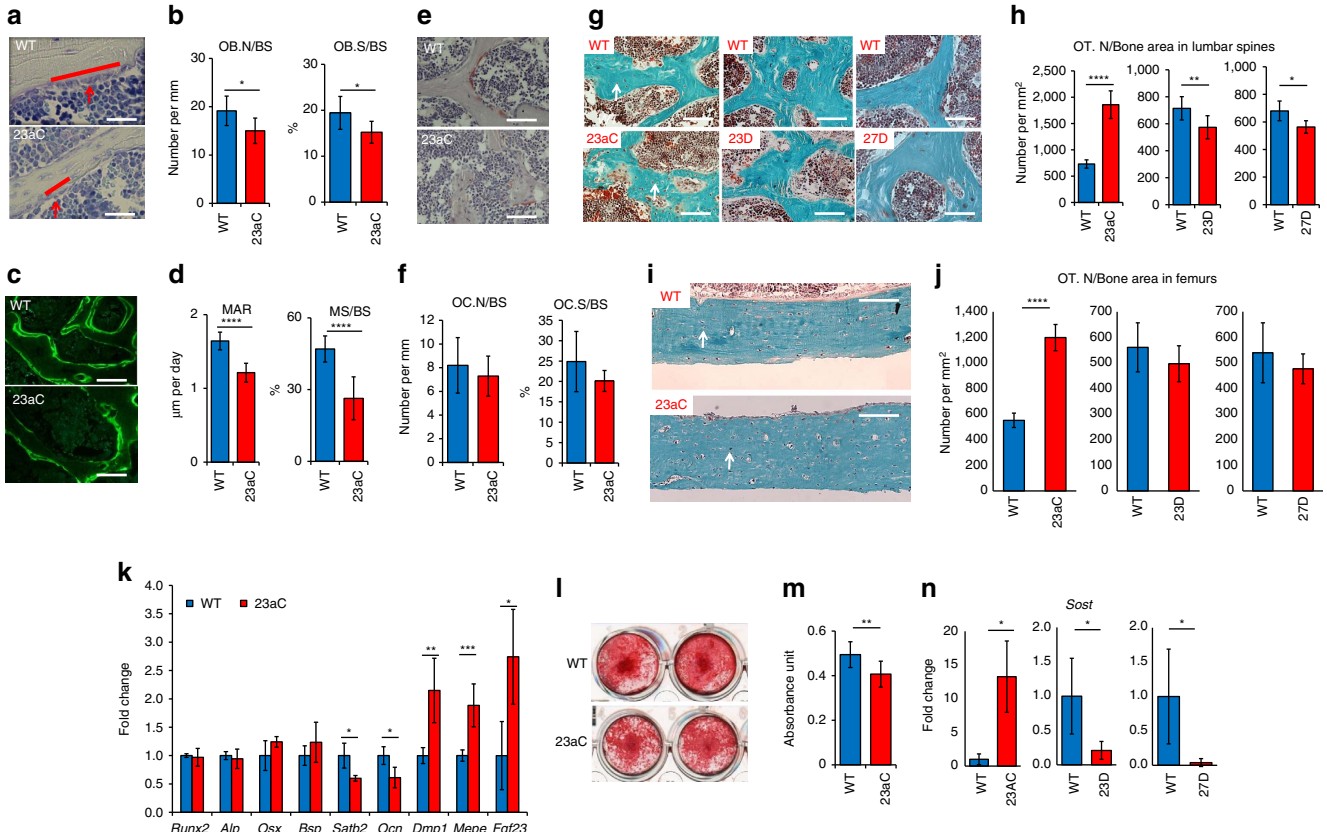

**Figure 2 | The miR-23a cluster regulates osteocyte differentiation.** (**a**) Toluidine blue-stained spine sections show osteoblasts (red arrows) on the trabecular bone surface (red lines). WT, wild-type; 23aC, *Col1a1-miR-23aC*. White bars indicate scale of 50 μm. (**b**) Osteoblast number/bone surface (OB.N/BS) and osteoblast surface/bone surface (OB.S/BS) are decreased in *Col1a1-miR-23aC* mice (N = 7). (**c**) Calcein double labelling shows the mineralization ability of osteoblasts on the trabecular bone of spines in *Col1a1-miR-23aC* mice. White bars indicate scale of 100 μm. (**d**) Mineral apposition rate (MAR) and mineralizing surface/bone surface (MS/BS) are decreased in *Col1a1-miR-23aC* mice (N = 7). (**e**) TRAP staining showing osteoclasts in *Col1a1-miR-23aC* mice. White bars indicate scale of 100 μm. (**f**) Osteoclast number/bone surface (OC.N/BS) and osteoclast surface/bone surface (OC.S/ BS) are not changed in *Col1a1-miR-23aC* mice (N = 7). (**g**) Trichrome staining of spines showing osteocytes (white arrows) in *Col1a1-miR-23aC, Col1a1-miR-23a* decoy and *Col1a1-miR-27a* decoy (27D) transgenic mice and WT littermates. White bars indicate scale of 100 μm. (**h**) Osteocyte number (OT.N/Bone area) was increased in *Col1a1-miR-23aC* mice but decreased in *Col1a1-miR-23a* decoy and *Col1a1-miR-27a* decoy mice (N = 7). (**i**) Representative images of trichrome staining showing osteocytes (white arrows) in cortical bones of femurs of Col1a1-miR-23aC mice and WT littermates. White bars indicate scale of 100 μm. (**j**) Osteocyte density was significantly increased in cortical bones of femurs of Col1a1-miR-23aC mice (N = 7) but not different in Col1a1-miR-23a decoy and (**d**) Col1a1-miR-27a decoy mice (N = 7 for each group). (**k**) The expression level of bone markers in P3 calvarial bones of WT and *Col1a1-miR-23aC* mice (N = 4). (**l**) Representative images of Alizarin Red staining of BMSC differentiation on day 14. WT, wild type; 23aC, *Col1a1-miR-23aC*. (**m**) Quantification of dissolved Alizarin Red from BMSCs shows decreased mineralization in *Col1a1-miR-23aC* mice (N = 9). (**n**) Expression level of *Sost* is significantly increased in cells from *Col1a1-miR-23aC* mice but significantly decreased in cells from *Col1a1-miR-23a* decoy and *Col1a1-miR-27a* decoy mice compared to WT controls after BMSC differentiation on day 28 (N = 3 for each). *t*-test was performed for the statistical analyses. Error bars represent the s.d. *P < 0.05, **P < 0.01, ***P < 0.005 and ****P < 0.001.

osteoblasts into osteocytes, we performed RNA-Seq analysis on RNA isolated from calvarial bones of *Col1a1-miR-23aC* mice and WT littermates (Supplementary Data 1). Forty-seven million reads out of 60 million reads per sample were mapped to the mouse genome, and a total of 300 genes were identified as being differentially expressed (P < 0.05, Supplementary Data 2); 20 of the most highly differentially expressed genes are listed in Supplementary Table 1. Using IPA (Fig. 3a), we discovered several affected pathways, and TGF-β signalling was significantly altered in the *Col1a1-miR-23aC* mice compared to their WT littermate controls (Supplementary Table 2). Specifically, downstream targets of TGF-β signalling such as *Hpgd* and *Cdkn1a* were significantly upregulated (Supplementary Data 2). IPA also revealed that *Prdm16* (also called *Mel1*), a TGF-β upstream regulator, was decreased in calvarial bones of *Col1a1-miR-23aC* mice (Fig. 3a and Supplementary Data 2). *Prdm16* was reported as a transcriptional co-repressor that inhibits TGF-β function by

recruiting HDAC1 to form a complex with Smad2/3, which leads to repression of TGF-β downstream genes[14,15]. More interestingly, the 3′UTR of *Prdm16* contains a miR-27a-binding site conserved across species (Fig. 3b and Supplementary Data 3), and Prdm16 has been shown to be regulated by miR-27a during adipogenesis[16]. To confirm that *Prdm16* is a direct target of miR-27a in osteoblastic cells, we performed a luciferase reporter assay by cloning the 3′UTR of *Prdm16* into the psiCHECK vector in HEK293T cells. Luciferase activity was decreased in the presence of ectopic miR-27a expression, and this suppression was relieved by expressing a miR-27a decoy (Fig. 3c). We further evaluated this effect in osteoblast lineage MC3T3-E1 cells. First, we generated stable MC3T3-E1 cell lines, which can be induced to express miR-27a or miR-27a decoy after treatment with doxycycline. After 24 h of treatment, the same luciferase reporter was transfected into the inducible MC3T3-E1 cells. Consistent with our previous data, induction of miR-27a

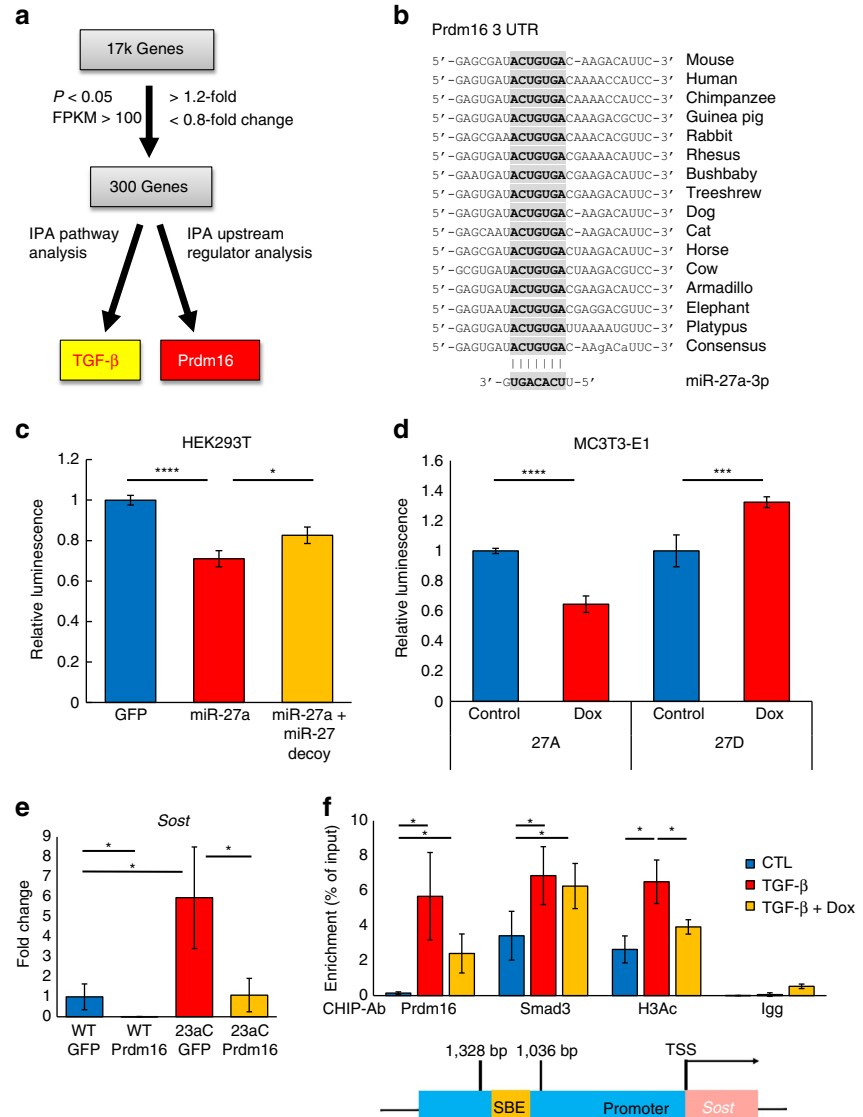

**Figure 3 | Prdm16 is a direct target of the miR-23a cluster and suppresses _Sost_.** (**a**) RNA-Seq with IPA and Targetscan analyses revealed that TGF-β is the most differentially regulated signalling pathway (P value = 4.75E − 06) and that Prdm16 is a possible repressive upstream regulator of TGF-β signalling (P value = 1.47E − 02). (**b**) The conserved seed sequence of miR-27a in the 3′UTR of _Prdm16_ mRNA among different species. (**c**) HEK293T cells were independently co-transfected with psiCheck-2 luciferase reporter containing 3′UTR _Prdm16_ along with the GFP control, miR-27a or miR-27a decoy (N = 3 for each group). (**d**) MC3T3-E1 cells were transduced with miR-27a (27A)- or miR-27a decoy (27D)-expressing lentivirus. Twenty-four hours before transfection of the psiCheck-2 luciferase reporter containing 3′UTR _Prdm16_, cells were treated with 1 μM of doxycycline (Dox) or vehicle (control); 48 h after transfection of the reporter vectors, cells were collected for luminescence signal reading (N = 4 for each group). (**e**) Expression levels of _Sost_ in BMSCs from WT or _Col1a1-miR-23aC_ (23aC) mice transduced with GFP- or _Prdm16_-expressing lentivirus on day 28 of differentiation (N = 3 for each group). (**f**) Prdm16 and Smad3 are co-localized at the promoter region (including one SBE) from 1,036 to 1,328 bp upstream of the _Sost_ transcription start site (TSS) in MC3T3-E1 cells. After 24 h of TGF-β treatment (10 ng ml$^{-1}$), acetylation of histone H3 (H3Ac) was increased with the recruitment of Smad3. Upon induction of _Prdm16_ (1 μM of doxycycline), H3Ac levels were decreased at the _Sost_ promoter region. CTL, control without Dox. This experiment was repeated three times. Statistical analyses used one-way ANOVA for multiple groups. Data are shown as mean ± s.d.; *P < 0.05, ***P < 0.005 and ****P < 0.001.

suppressed _Prdm16_ reporter activity in the stable MC3T3-E1 cell lines (Fig. 3d). These data support that _Prdm16_ is a direct target of miR-27a and can be regulated post-transcriptionally in osteoblast lineage cells.

_Prdm16_ is a well-known regulator of brown fat determination, palatogenesis, neural and haematopoietic stem cell maintenance[17–22], but its role in osteogenesis is poorly understood. To test whether _Prdm16_ affects the differentiation of osteoblasts into osteocytes, we transduced BMSCs from _Col1a1-miR-23aC_ transgenic and WT mice with a pInducer20 lentivirus[23] expressing green fluorescent protein (GFP) or _Prdm16_ (Supplementary Fig. 10b) and measured changes in

expression of the osteocyte marker _Sost_. Interestingly, the elevated _Sost_ expression observed in differentiating _Col1a1-miR-23aC_ BMSCs was rescued to the WT levels by overexpressing _Prdm16_ (Fig. 3e). Taken together with the data of direct suppression of _Prdm16_ by miR-27a shown in Fig. 3c,d, we conclude that _Prdm16_ is a negative regulator of osteoblast terminal differentiation into osteocytes, which is controlled post-transcriptionally by the miR-23a cluster.

To assess how _Prdm16_ inhibits TGF-β signalling during terminal osteogenic differentiation, a chromatin immunoprecipitation (ChIP) assay was performed in MC3T3-E1 cells engineered to stably overexpress _Prdm16_ after treatment with doxycycline

(Supplementary Fig. 10c). In the presence of TGF-β, both Prdm16 and Smad3 occupied the endogenous *Sost* promoter region compared to the 3′UTR region of *Gapdh*, which was the negative control (Fig. 3f and Supplementary Fig. 12). However, histone H3 acetylation (H3K9Ac and H3K14Ac), the hallmark of active transcription, was also enriched at this site (Fig. 3f) in association with increased *Sost* gene expression (Supplementary Fig. 10d). Upon doxycycline treatment and induction of Prdm16, histone H3 acetylation was decreased in association with decreased *Sost* expression (Fig. 3f and Supplementary Fig. 10d). This repressive function of Prdm16 could be mediated via HDAC1 recruitment, as has been seen for other systems[15]. We also observed a trend of decreased Prdm16 occupancy with TGF-β and doxycycline although neither this nor Smad3 occupancy was statistically different from occupancy in presence of TGF-β only (Fig. 3f). One possible explanation may be squelching of other transcriptional repressors by Prdm16. Taken together, these data show for the first time that Prdm16 is a negative epigenetic regulator of TGF-β signalling in its inhibition of terminal osteoblast differentiation into osteocytes in part by repressing expression of the osteocyte-specific gene *Sost*.

**TGF-β signalling is dysregulated in GOF mice.** The function of TGF-β signalling in osteocytes was further validated *in vivo* by crossing *Col1a1-miR-23aC* mice with TGF-β reporter mice. This reporter mouse line carries a synthetic reporter gene containing 12 repeats of Smad-binding elements (SBE) fused to *luciferase* and responds to Smad2/3-dependent TGF-β signalling *in vivo*[24]. The *Col1a1-miR-23aC* GOF reporter mice showed elevated luciferase activity compared to WT littermates, indicating increased TGF-β activity *in vivo* in this mouse model (Fig. 4a,b). By contrast, the *Col1a1-miR-23a* decoy and *Col1a1-miR-27a* decoy LOF reporter mice showed decreased bioluminescence (Fig. 4a,b). Indeed, this *in vivo* study indicates that the miR-23a cluster regulates the TGF-β signalling pathway *in vivo* during bone homeostasis.

To test whether TGF-β is the dominant signalling pathway contributing to the elevated osteocyte density and decreased bone mass seen in the *Col1a1-miR-23aC* mice, we treated these mice with an anti-TGF-β1 antibody (1D11) or control antibody (13C4). The low bone mass phenotype was rescued by 1D11 to a level comparable with that seen in WT mice treated with 13C4 (Fig. 4c). Bone histomorphometric analysis showed decreased osteocyte density in *Col1a1-miR-23aC* mice after 1D11 treatment (Fig. 4d,e) and decreased osteoclast number and surface (Supplementary Fig. 13a,b) compared to *Col1a1- miR-23aC* mice after treatment with control antibody. Furthermore, there was a trend for increased mineral apposition rate and mineralizing surface (Supplementary Fig. 13c,d), without significant change in osteoblast number and surface following 1D11 treatment (Supplementary Fig. 14). These histomorphometric data suggest that, as previously reported, TGF-β signalling affects different aspects of bone homeostasis leading to the combination of increased bone formation and decreased bone resorption in *Col1a1-miR-23aC* mice.

In the recent decades, the functions of osteocytes and the factors that regulate their differentiation have been the focus of several *in vitro* cell studies and a few genetic mouse models[25–27]. So far, no miRNA has been proven to regulate terminal differentiation of osteoblasts into osteocytes *in vivo*. Few *in vitro* studies have suggested that several osteocyte-specific miRNAs may be important[28]. Here we show that the miR-23a cluster regulates osteoblast-to-osteocyte differentiation using both GOF and LOF mouse models. Prior cell studies showed that the miR-23a cluster was induced during differentiation and could

regulate osteoblast differentiation *in vitro*[7,29]. Our study demonstrates a specific role for the miR-23a cluster in osteocyte differentiation *in vivo*, but also identifies how it targets TGF-β signalling function during this process. Elevated TGF-β signalling has been shown to increase osteocyte density in different mouse models and in the patients and mice with Osteogenesis Imperfecta[30,31], supporting its physiologically important role in osteocyte determination. Our study further emphasizes the function of TGF-β signalling in osteocytes by revealing an upstream regulatory factor, *Prdm16*, directly regulated by miR-27a. Prdm16 is known to control brown fat cell identity and is required for normal palatogenesis[19,21]. Moreover, we now uncover a novel function of the Prdm16/TGF-β signalling nexus during the terminal differentiation of osteoblasts into osteocytes in part by repression of osteocyte-specific *Sost* expression. At the early stage of osteogenesis, Prdm16 is expressed and forms an inhibitory complex with Smad2/3 to suppress the downstream targets of TGF-β (Fig. 5a). At later stages, the increased expression of the miR-23a cluster can directly downregulate *Prdm16* by targeting its 3′UTR, which relieves the repression of the downstream targets of TGF-β signalling in promoting the differentiation of osteoblasts into osteocytes (Fig. 5a,b). Overall, our study has identified a physiological role for the miR-23a cluster in the regulation of the terminal differentiation of osteoblasts into osteocytes via Prdm16/ TGF-β signalling (Fig. 5b). As such, it serves as a potential novel target for controlling osteocyte number in bone diseases.

## Methods

**Mice.** To generate the osteoblast-specific GOF *Col1a1-miR23aC* transgenic mice, a 991 bp fragment of genomic DNA containing the miR-23a cluster was cloned downstream of the 2.3 kb Collagen type I, alpha 1 (Col1a1) 2.3 kb promoter into a transgenic vector[6] containing the tyrosinase minigene and the woodchuck post-transcriptional regulatory element sequence (Supplementary Fig. 1a). To generate the osteoblast-specific LOF *Col1a1-miR-23a* decoy, *Col1a1-miR-27a* decoy and *Col1a1-miR-24-2* decoy transgenic mice, 368 bp DNA fragments with nine decoy repeats for each miRNA[12,13] were synthesized at BlueHeron and cloned into the Col1a1 promoter transgenic vector. Transgenic founders were generated by pronuclear injections in FVB/N embryos. Transgenic mice were identified by eye and fur pigmentation, and the genotypes were confirmed by PCR using primers specific for the woodchuck posttranscriptional regulatory element sequence. TGF-β reporter mice expressing luciferase in response to the Smad2/3-dependent TGF-β signalling pathway (SBE-luc mice) were obtained from The Jackson Laboratory (B6.Cg-Tg(SBE/TK-luc)7Twc/J) and bred to *Col1a1-miR23aC*, *Col1a1-miR-23a* decoy and *Col1a1-miR-27a* decoy transgenic mice to generate the GOF and LOF mice expressing the TGF-β reporter transgene and wild-type littermates. These studies were approved by the Baylor College of Medicine Institutional Animal Care and Use Committee.

**Anti-TGF-β treatment.** We treated 6-week-old female *Col1a1-miR23aC* mice and WT littermates with the pan-TGF-β neutralizing antibody (Genzyme; clone 1D11) for 6 weeks (10 mg per kg body weight, intraperitoneally (i.p.), three injections per week). Control groups of *Col1a1-miR23aC* mice and WT littermates were treated with an control antibody (Genzyme; clone 13C4) of the same IgG1 isotype. Mice of each litter were randomly assigned to treatment (1D11) or control groups (13C4). After treatment, mice were killed and lumbar spines were collected and fixed in 10% formalin for μCT and bone histomorphometry.

**μCT analysis.** The bone samples were harvested from 3-month-old female and male mice. Lumbar vertebrae, femurs and skulls were placed into a 16 mm tube filled with 70% ethanol and scanned by a Scanco μCT-40 desktop microCT scanner (Scanco Medical) for quantification of bone parameters. The vertebral and femoral trabecular bone parameters were obtained, using the Scanco Analysis Software, from the vertebral body L4 and the distal metaphyseal section of the femur. The operators were blinded to conduct the analyses.

**Histomorphometric analysis.** The bone samples were harvested from 3-month-old female mice. We performed toluidine blue, trichrome and TRAP staining of undecalcified lumbar vertebrae plastic sections with 5–7 mm thickness. The static and dynamic histomorphometric analysis was performed by calcein injection (Sigma-Aldrich, 15 mg per kg body weight) with an interval of 4 days after the first injection, and data were analysed with the Bioquant Osteo Image Analysis System. The operators were blinded to conduct all histomorphometric analyses.

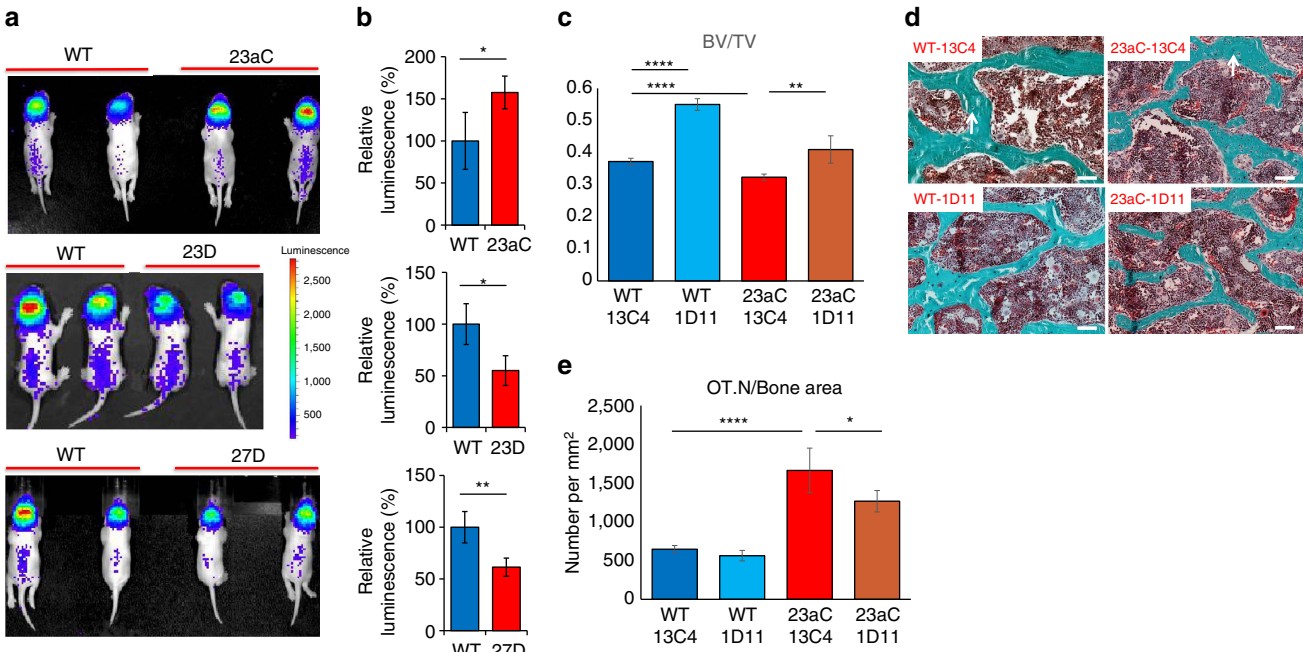

**Figure 4 | Dysregulation of TGF-β signalling drives the phenotype in GOF mice.** (**a**) Representative luminescence images of TGF-β reporter mice (P3) harbouring different transgenes. TGF-β reporter activity in the head is from active bone formation in the postnatal growing cranium. 23aC, *Col1a1-miR-23aC*; 23D, *Col1a1-miR-23a decoy*; 27D, *Col1a1-miR-27a decoy*. (**b**) Luminescence quantification shows increased activity of TGF-β signalling in *Col1a1-miR-23aC* (23aC) mice and decreased activity in *Col1a1-miR-23a decoy* (23D) and *Col1a1-miR-27a decoy* (27D) mice ($N = 4$ for each group). (**c**) μCT analysis of BV/TV in lumbar spines. The low bone mass phenotype in *Col1a1-miR-23aC* mice was rescued by anti-TGF-β antibody treatment (23aC 1D11) to levels comparable with those in WT mice treated with control antibody (WT13C4; 12-week old females; $N = 4$ for each group). (**d**) Representative trichrome staining images of spines showing osteocyte (white arrows) density in Col1a1-miR-23aC (23aC) and WT littermates treated with 1D11 or 13C4. White bars indicate scale of 100 μm. (**e**) Osteocyte density (OT.N/Bone area) measured by bone histomorphometric analysis. Osteocyte density in *Col1a1-miR-23aC* mice was reduced by 1D11 (23aC 1D11) compared to 13C4 (23aC 13C4; $N = 4$ for each group). Statistical analyses used *t*-test for paired groups and one-way ANOVA for multiple groups. Data are shown as mean ± s.d.; *$P < 0.05$, **$P < 0.01$, ****$P < 0.001$.

**In vivo bioluminescence imaging.** We injected P3 transgenic *Col1a1-miR23aC*, *Col1a1-miR-23a* decoy, *Col1a1-miR-27a* decoy mice and WT littermates expressing the TGF-β reporter transgene with D-luciferin (Gold Bio, 150 mg kg$^{-1}$, i.p.), anaesthetized them with isoflurane (Piramal) and performed imaging 10 min after injection using a bioluminescence imaging system (Xenogen).

**Cell culture and osteogenic differentiation.** HEK293T and MC3T3-E1 cells (from American Type Culture Collection) were cultured in DMEM (Hyclone) or α-MEM medium (Hyclone), respectively, with 10% fetal bovine serum (Gibco), 1% glutamine (Gibco) and 1% penicillin–streptomycin (Gibco). Bone marrow cells were isolated from tibiae and femurs of ∼2-month-old male mice and cultured in α-MEM medium with 10% fetal bovine serum, 1% glutamine and 1% penicillin–streptomycin. The medium was changed every 2 days and unattached cells were removed. After 7 days, we reseeded the attached cells, defined as BMSCs. BMSCs were seeded at $2.5 \times 10^4$ cells per cm$^2$ in 24-well plates (for Alizarin Red staining) or 6-well plates (for RNA extraction). To induce osteogenesis, BMSCs were cultured with additional ascorbic acid (Sigma-Aldrich, 100 μg ml$^{-1}$) and β-glycerophosphate (Sigma-Aldrich, 5 mM) for 14 days (Alizarin Red staining) or for 28 days (qRT–PCR).

**Alizarin Red staining.** After being seeded into 24-well plates, BMSC osteogenesis was induced for 14 days. Cells were fixed in 4% paraformaldehyde (Sigma-Aldrich) and stained with Alizarin Red S (Sigma-Aldrich). To quantify the mineral material in the culture, the stain was dissolved with 10% cetylpyridinium chloride, and the absorbance was measured with the FLUOstar OPTIMA Microplate Reader (BMG LABTECH).

**Lentivirus transduction.** Prdm16 cDNA (from Cell-Based Assay Screening Service Clone at Baylor College of Medicine, clone ID: 9335094) was cloned into the pINDUCER20 lentiviral vector[23] (from Dr Trey Westbrook). The lentiviruses (pINDUCER20-Prdm16) were produced with packaging plasmids pMD2.G and psPAX2 (from Dr Didier Trono) in HEK293T cells. MC3T3-E1 cells and BMSCs were transduced by spin-infection for 30 min at 1,000*g*. Transduced MC3T3-E1 cells were selected by using G-418 (ThermoFisher Scientific) at 250 μg ml$^{-1}$. To

induce expression of Prdm16, cells were cultured in the medium with 1 μM doxycycline (Sigma-Aldrich).

**Luciferase reporter assay.** A 1,516 bp DNA fragment with the predicted miR-27a-binding site (Fig. 3b) from the 3′UTR of Prdm16 was cloned from FVB/N mouse genomic DNA to the multiple cloning regions of the luciferase reporter psiCHECK-2 vector (Promega). A 236 bp fragment of genomic DNA containing miRNA miR-27a was cloned into the pLKO.1 vector (addgene) to express miR-27a; the 368 bp miRNA miR-27a decoy DNA fragment synthesized at BlueHeron was cloned into the pLKO.1 vector to express the miR-27a decoy. HEK293T cells were co-transfected with the luciferase reporter vector and the vector expressing miR-27a, miR-27a decoy or GFP control. After 48 h, the cells were harvested and the luciferase signals were measured with the Dual-Luciferase Reporter Assay System (Promega).

**qRT–PCR.** Total RNA was extracted from either bone tissues of mice or cultured cells with TRIzol reagent (Invitrogen). cDNAs were synthesized from extracted RNA with the Superscript III First Strand RT–PCR Kit (Invitrogen), and real-time quantitative PCR amplifications were performed in a LightCycler (Roche). β2-microglobulin was used as the internal control to normalize gene expression. For microRNA qRT–PCR, TaqMan MicroRNA Assays (Applied Biosystems) were used to quantify the expression of mature miR-23a (Assay ID: 000399), miR-27a (Assay ID: 000408) and miR-24-2 (Assay ID: 002494). Amplication and detection were performed using the 7500HT Fast Real-Time PCR system (Applied Biosystems) and the TaqMan Universal PCR Master Mix. The expression of mature miRNAs was normalized with internal control sno202 RNA (Assay ID: 001232).

**RNA-Seq.** Total RNA was extracted by the same method used for qRT–PCR. Primary miRNAs and mRNAs were captured by Dynabeads Oligo (dT)$_{25}$ magnetic beads (Invitrogen) and fragmented by using the NEBNext Magnesium RNA Fragmentation Module (New England Biolabs). Double-stranded cDNAs were synthesized with the SuperScript Double-Stranded cDNA Synthesis Kit (Invitrogen) and used as the library template. Libraries were generated with the TruSeq RNA Library Preparation Kit (Illumina). Sequencing was performed on an Illumina HiSeq 2,000 instrument as 100 bp pair-end reads at the Human Genome

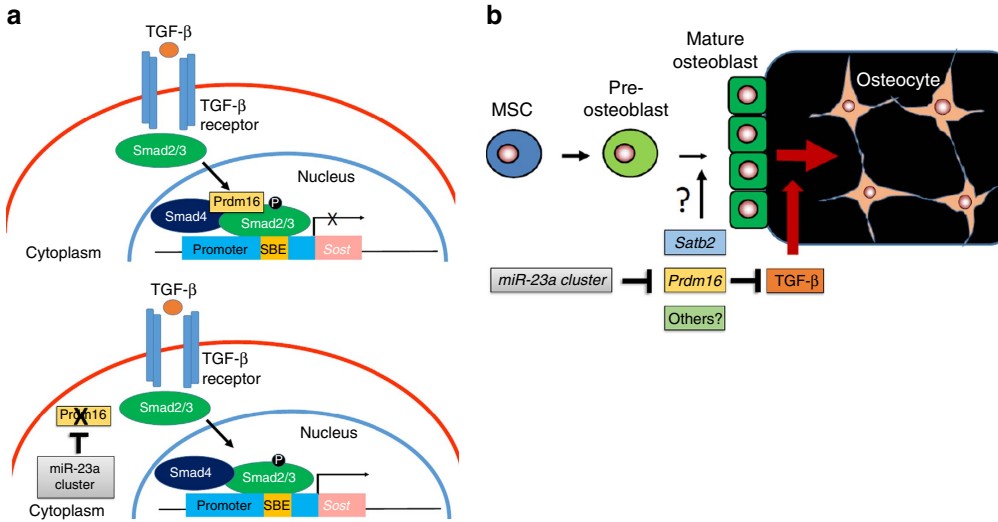

**Figure 5 | The models of miR-23a cluster in osteocyte differentiation.** (**a**) A schematic model of miR-23a cluster/Prdm16/TGF-β signalling axis. In the absence of miR-23a cluster, Prdm16 forms a negative complex with phosphorylated Smad2/3 and is recruited to the promoter of Smad-binding element (SBE) to suppress downstream target genes such as *Sost*. When miR-23a cluster is upregulated during osteoblast differentiation, Prdm16 is suppressed by the miR-23a cluster in a direct manner, which leads to increased H3 acetylation in the regulatory region, followed by induction of *Sost* by the phosphorylated Smad2/3 complex. (**b**) A model of regulation of osteocyte differentiation by the miR-23a cluster. During mature osteoblast to osteocyte differentiation, the miR-23a cluster suppresses *Prdm16*, a negative regulator of TGF-β signalling and enhances TGF-β signalling to accelerate osteocyte differentiation. The miR-23a cluster also downregulates other targets, such as *Satb2*, one of the osteoblast transcription factors. The role of the miR-23a cluster at the early stages of osteoblast differentiation is not clear yet. MSC, mesenchymal stem cell.

Sequencing Center at Baylor College of Medicine[30]. Raw reads were mapped to mouse genome mm9 and splice junction sites with bowtie (v0.12.7) and Tophat (v2.0.0) in the strand-specific model when the strand-specific library protocol was applied. The reference Mouse genome annotation file was downloaded from UCSC (http://genome.ucsc.edu/) with a timestamp of 30 August 2011. Read counts mapped to each gene were calculated by HTseq (http://www-huber.embl.de) with the default model. Fragments per kilobase of exon model per million fragments mapped values were calculated using Cufflinks (version 2.1.1, http://cufflinks.cbcb.umd.edu). Differential expression was analysed with R (v2.14.0, http://www.R-project.org) and Bioconductor (release 2.10) with the R package DESeq (v1.6.0). Pathways and upstream regulator analyses were performed by IPA (http://www.ingenuity.com/) with default parameters. miRNA targets were predicted by Targetscan (http://www.targetscan.org/).

**ChIP–PCR assay.** MC3T3-E1 cells were transduced with the pINDUCER20-Prdm16 virus and selected with G-418 to generate cells with inducible expression of Prdm16. Cells were maintained in the medium without doxycycline. Cells were seeded at $5 \times 10^4$ cells per cm$^2$ into 20 mm Petri dishes with or without 1 μM doxycycline. Twenty-four hours later, the medium was changed to a serum-free medium with or without 1 μM doxycycline, and 24 h later the medium was changed to a serum-free medium with or without 1 μM doxycycline and 10 ng μl$^{-1}$ TGF-β1 (R&D Systems). Finally, 24 h later, the cells were collected with Magna ChIP A/G One-Day Chromatin Immunoprecipitation Kits (Millipore). The sheared chromatin was immunoprecipitated with 10 μg ml$^{-1}$ anti-Prdm16 antibody (R&D Sysytems, AF6295), 10 μg ml$^{-1}$ anti-Smad3 antibody (Abcam, ab28379), 10 μg ml$^{-1}$ anti-Acetyl-Histone H3 antibody (Millipore, 17–615) or 10 μg ml$^{-1}$ anti-IgG antibody (Millipore, PP64-1KC). DNA was quantified by qRT–PCR with different primer sets and normalized with input DNA of each group. The 2 kb upstream region from the mouse *Sost* transcription start site was divided into six regions (S1–S6). Only the S3 region (Fig. 3f,1,036–1,328 bp upstream from transcription start site) exhibited significantly difference among treatments.

**Statistical analysis.** Statistical analyses on groups of two were performed using the Student's *t*-test, and a *P* value < 0.05 was considered to be statistically significant. Groups of three or more were analysed by one-way ANOVA, and the Tukey's HSD test was used when *P* value was < 0.05. Data are shown as mean ± s.d.; *$P$ < 0.05, **$P$ < 0.01, ***$P$ < 0.005, ****$P$ < 0.001.

**Data availability.** Sequence data that support the findings of this study have been deposited in the National Center for Biotechnology Information Sequence Read Archive (SRA) database with the primary accession code SRP095626. Other data that support the findings of this study are available from the corresponding authors.

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

## Acknowledgements

We thank Drs Patrick Finn and Steven Ledbetter from the Genzyme Research Center (Framingham, Massachusetts, USA) for providing the anti-TGF-β and control antibodies for our study. In addition, we also thank Dr Patricia Fonseca for editorial assistance, Megan Bagos and Carrie Jiang for assistance with the µCT analyses. This work was supported by the BCM Intellectual and Developmental Disabilities Research Center (HD024064) from the Eunice Kennedy Shriver National Institute of Child Health and Human Development, the BCM Advanced Technology Cores with funding from the NIH (AI036211, CA125123 and RR024574), the Rolanette and Berdon Lawrence Bone Disease Program of Texas and the BCM Center for Skeletal Medicine and Biology. This work was supported by the NIH (HD70394 to B.H.L.).

## Author contributions

H.-C.Z. performed most of the experiments. H.-C.Z., Y.B., J.T. and B.H.L. designed the studies. B.C.D., Y.C., T.B. and E.M. provided technical support. P.M.C. assisted with decoy design. R.C. performed the RNA-Seq experiments and analysis. H.-C.Z., Y.B. and B.H.L. wrote the manuscript.

## Additional information

**Competing interests:** The authors declare no competing financial interests.

**How to cite this article**: Zeng, H.-C. *et al.* MicroRNA miR-23a cluster promotes osteocyte differentiation by regulating TGF-β signalling in osteoblasts. *Nat. Commun.* **8**, 15000 doi: 10.1038/ncomms15000 (2017).

**Publisher's note**: 

