## [Peer Review File · Nature Communications]

Reviewers' comments:

Reviewer #1 (Remarks to the Author):

This study finds that gain-of-function (GOF) and loss-of-function miR-23a transgenes have opposing effects on osteocyte density. The authors then attempt to link these changes to alterations in Prdm16, which is a negative regulator of TGFbeta signaling. While many of the results are interesting, they are in general over-interpreted. Additional studies are required to support the conclusions drawn by the authors, especially the models proposed in figure 4.

1. The tissue-specificity of each transgene that has an effect on bone mass needs to be verified. Along the same lines, if multiple independent lines were generated for each transgenes, the authors need to indicate if the bone phenotype was present in each line.
2. Both GOF and LOF transgenes resulted in low bone mass but had different effects on osteoblast number and osteocyte density. However, no explanations or possible mechanisms responsible for these discordant results were provided or discussed. Along the same lines, the results shown in figure 2i demonstrate that the GOF and LOF transgenes have opposite effects on Sost mRNA levels, at least in vitro. If similar levels of regulation occur in vivo, one would have expected opposing effects on osteoblast number or bone formation when comparing the GOF and LOF mice. However, this was not the case. Some explanation or discussion of this lack of effect is required.
3. The authors suggest that Sost is an important indirect target of the miRNAs mentioned in this study. However, Sost mRNA measurements were exclusively from cultured cells. Sost mRNA levels in bones tissue from the transgenic mice needs to be measured to confirm the in vitro results.
4. Structural and histological analyses of the skeleton are overly limited. The effects of the transgenes on cortical bone mass should be included as well as the osteocyte density in cortical bone, at least for the GOF mice.
5. On page 7, the authors conclude that the changes in Sost mRNA contribute to the changes in osteocyte density. However, Sost KO mice do not display a change in osteocyte density, at least in cortical bone (JBMR 24:1651, 2009). Therefore, their conclusion is not supported by experimental evidence.
6. Discussion of the ChIP results shown in figure 3e is misleading. In contrast to what is stated by the authors, over-expression of Prdm16 does not lead to more occupancy of the Sost locus by this factor but instead either no change or perhaps lower levels. Therefore, the conclusion that Prdm16 is directly responsible for the reduced histone acetylation and reduced Sost expression is not supported by these results. Moreover, ChIP results from a control region or control locus should be presented in the extended data section.
7. Inhibition of TGFb signaling in vivo increased bone volume and reduced osteocyte density, but did not alter osteoblast number. There is no explanation for how this maneuver increased bone mass. The authors need to address how this antibody affected their proposed target of TGFb signaling, Sost, in vivo. Moreover, if neither bone formation nor Sost levels were affected by the antibody, then their model shown in figure 5e does not accurately reflect their results and should be modified.
8. A section in the methods should be included to describe the statistical analysis.
9. A minor point is that it is unclear in the methods section what the 5-7 mm refers to when describing the histological analysis. Is it tissue size, section thickness, or something else?

Reviewer #2 (Remarks to the Author):

Osteocytes are the terminally differentiated cell type of the osteoblastic lineage. Osteocytes serve important functions in skeletal homeostasis including controlling osteoclastogenesis and biomechanical sensing. Although the transcriptional regulation of osteoblast differentiation has been well-characterized, the factors that regulate differentiation of osteocytes from mature osteoblasts are poorly understood.

Here, the authors show that the microRNA cluster miR-23a~27a~24-2 promotes osteocyte differentiation with osteoblast-specific gain-of-function (Col1a1-miR23aC) mice showing low bone mass associated with decreased osteoblasts but increased osteocyte numbers whilst loss-of-function transgenic mice overexpressing microRNA decoys for either miR-23a or miR-27a, but not miR24-2, showed decreased osteocyte numbers. RNA-sequencing analysis of bones from the gain-of-function Col1a1-miR23aC transgenic mice showed altered transforming growth factor β (TGF- β) signaling involving Prdm16 transcription factor, a negative regulator of the TGF- β pathway was involved in the regulation of osteoblast differentiation. Prdm16 was directly repressed by the miR-23a cluster with concomitant alteration of Sclerostin (Sost) expression in osteocytes. Inhibition of TGF- β activity using neutralizing antibody, 1D11, rescued the low bone mass and osteocyte phenotypes observed in the gain-of-function Col1a1-miR23aC transgenic mice. Taken together, the miR-23a cluster regulates osteocyte differentiation by modulating the TGF- β signaling pathway through targeting of Prdm16.

This manuscript is well written and clearly structured. Experimental design is excellent with a novel but clear proposed mechanism and appropriate models used to validate findings and statistics appropriately applied. Figures are generally of high quality and presented in a logical fashion and conclusions drawn are appropriate to the interpretation of results in terms of the experiments conducted. The manuscript outlines the role of miR23a cluster in the regulation of terminal differentiation of osteoblasts to osteocytes via a mechanism involving TGF beta and its upstream regulator Prdm16.

There are, however some few points of consideration that could generally help improve the paper.

1. The authors to further outline the central role of TGF beta signaling pathway in the increase in osteocyte numbers in the Col1a1-miR-23aC mice, treated these mice with an anti-TGF- β 1 antibody (1D11) or control antibody (13C4). The low bone mass phenotype was rescued by 1D11 to levels comparable with those seen in WT mice treated with 13C4 (Fig. 4c). Bone histomorphometric analysis showed decreased osteocyte density in Col1a1- miR-23aC mice after 1D11 treatment (Fig. 4d); however, no significant change in osteoblast number was found following 1D11 treatment (Extended Data Fig. 5). Would have been interesting to see if these changes in osteocyte numbers(decrease) and bone mass phenotype were also observed in Col1a1- miR-23aC mice with specific TGF beta knockout in osteoblast.

2. There is no list of keywords.

Reviewer #3 (Remarks to the Author):

Nature Communication Review

Criteria:

Major claims: novel, of interest, conclusions original, if not provide refs

What additional is needed

Subjective- will it influence the field

Summary:

This manuscript describes the functional activities of the miR-23a-27a-24-2 cluster related to regulation of bone homeostasis by osteocytes. There are numerous papers in the literature (~22) documenting the effects of the cluster in vitro regulating ESCs, several hematopoietic lineages and

osteoblast, muscle differentiation. Using a transgenic model expressing the entire cluster under control of the Col1a1 promoter for examining functional effects in osteoprogenitors and developing osteoblasts, the authors observed a phenotype of low bone mass, decreased osteoblasts and increased osteocytes, and was interpreted as the cluster promoting osteocyte differentiation. Inhibition of each of the miRNAs in the cluster was carried by a decoy strategy that also resulted in low bone mass mediated by miRNAs -23a and -27a; however, miR-24-2 which was the highest expressed (shown in extended data), had no LOF effect. The manuscript then goes on to characterize a mechanism focusing on the highest detected pathway, Prdm 16 -TGFB, by transcriptome RNA-Seq analysis between WT and tg GOF mouse. This is not novel as the linkage of this cluster to TGFB signaling has been reported in several studies in different cell types and (see e.g Chhabra review, PMID20815877). Further a Prdm-1 - TGFb1 axis, has also been previously described. Nonetheless much knowledge can be gained from in vivo studies and particularly for the first time in bone. The authors have generated two mouse models, over-expressing the miR-cluster and inhibiting each miRNA by a decoy method, but these are not fully characterized and indicate complex of systemic effects on bone mass, due likely to potential non-specific /off target effects. Although the Col1a1 2.3 is reasonably bone specific, any cell population expressing high levels of the cluster can potentially secrete the miRNA cluster and can effect non-osseous cells. Thus the findings are not readily interpretable.

Will this study be of general interest and influence the bone field - is questionable as the conclusion of the manuscript that inhibition of osteoblasts, yet more osteocytes. in the differentiation of osteocytes, is difficult to comprehend with fundamental biology that osteocytes form from surface osteoblasts. If TGFB is contributing to the phenotype additional evidence that for either the osteoblast to osteocyte conversion, or the direct phenotypic changes that occur in the osteocytes form the. While many experiments have been performed, and although they can explain, in part loss of bone, overall this study is not sufficiently rigorous to have a significant impact due to problems with the mouse models, data inconsistencies and interpretations which would be misleading without additional studies and detailed explanations of alternative interpretation.

One of the problems with presentation of the study is not having the benefit of adequate discussion of the data because of the format/style of Nature COMM publications.

Major concerns:

1. Related to non-specific effects in the tg mouse. The average FPKM for the miR-23a cluster is around 33 fragment per kilobase per millions. In extended data Figure 1d showing a range of 2.5 to 4.5-fold overexpression for the cluster members in mature osteoblasts. Therefore, the overall overexpression will be very broad and repress all possible, both highly and poorly conserved targets (approximately 300 genes differentially expressed in your study).
2. There are numerous gaps in data where comparisons cannot be made and inconsistencies that are not explained in the paper. Examples: The extended Figure 2 shows in GOF in females at 3-month very with very little change in BV/TV trabecular bone in femur when compared to main Fig 1c spine data. Specific LOF of miR-23a and miR 27a in extended Fig 2 does not show BV/TV for comparisons to spine in Fig 1e.
3. The absence of any analyses of cortical bone in the femur is a major deficiency in the study given that this is the major osteocyte tissue. The periosteal surface would be very informative related to osteocyte "differentiation" that can be analyzed in absence of the bone marrow environment
4. Fig 2c micrograph requires far more detail to explain the interpreted phenotype. More evidence is needed to determine if there is a conversion from OBs to OTs which is higher in miR-23aC model. At higher magnification can pre osteocyte be seen with some extension beginning the transition into a lacunar osteocyte in mineralized matrix. It would be far more informative to show

calcein labeling in long bone with rates on periosteal and endosteal surfaces The irregularity of the calcein labeling on trabecular surfaces is due to osteoclast activity.

Biologically osteocytes must originate from osteoblasts, unless there is some magical way more osteocytes (Fig. 2h).

5. Inhibitors of mineralization. True to osteocyte, that inhibitors on mineral deposition are increased in membranous bone of PN d3 mice (Fig 2i; but in figure 2j BMSC from GOF mice show a bit less mineralization. But the more revealing biological information is that osteoblast differentiation is NOT inhibited in BMSCs, if there is significant mineralization detected suggesting that the striking loss of bone in vivo is due to systemic effects and not cell autonomous effects due to miR cluster expression. More clarity in the phenotype would be very much appreciated by analysis of the early markers of the osteoblast, e.g collagen, alkphos, or some early transcription factors known to promote differentiation than is presented in extended Fig. 4a for osteocalcin only.

6. The decoy mice - more information is needed. Images of their bone (uCT) should be shown. Evidence that the decoy is working is needed. This reviewer does not have expertise with the procedure and only Refs 10, 11 are provided. Is the decoy targeting the cytosolic precursor or the mature miRNA? No confirmatory evidence is shown that the miR-decoys actually bind and inhibit the expression of miR-23a or 27a.

7. Fig 3. Bioinformatics and Prdm-TGFB axis. Panel a: How robust is this analysis? Are the data representative of multiple experiments or from long bones? Was this data compared to RNA seq from miR-23a or -27a decoy mice? A rationale that Prdm16 is the one among all the targets potentially carrying over the miR-23a cluster regulation in osteocytes should be provided. Was Prdm16 the only one that was detected by RNA-Seq? Supplementary data showing of the top 10-20 most highly differentially expressed (up and down regulated) and the top 5 most enriched pathways would be desirable information.

Figure 3. Cell studies are performed in 3 different cell lines. Panel c should have used MC3T3 osteoblasts for the reporter assay and panel d use BMSC (non-differentiated osteoblasts) which should be compared to MC3T3 osteoblasts that were used for the ChIP studies. Alternatively the PN3 calvarial bone (representing largely an osteocytes) from WT are TG mice are endogenous models of decreased and increased Prdm, respectively, not requiring transfection, and the phenotype could be rescued by TGFB antibody. Performing Q-PCR for the levels of TGFB, Prdm16, miRNA levels and osteocyte markers would contribute to the understanding functional relationships.

Panel 3c begs the question -What happens to the Prdm16 UTR LUC activity if the reporter is transfected into WT and 27a decoy cells? As a control, responsiveness with mutant Prdm16 UTR would be appropriate.

Panel 3d is a crucial experiment to establish a direct link between miR-23a cluster and Prdm16. The experiment lacks several controls. Instead of Prdm16 overexpression, Prdm16 shRNA will be more appropriate. Also to prove miR-23aC direct regulation, instead of overexpressing Prdm16 cDNA, it is necessary to overexpress Prdm16 cDNA {plus minus} 3'UTR. Since miRNAs regulate translation first, then mRNA, a western blot for Prdm16 could be shown as evidence direct regulation.

8. Fig 4 TGFB reporter mice contribute to identifying a mechanism that involves the miR cluster, bone loss due to TGFB inhibition and increased SOST that inhibits osteoprogenitor cells. If the miR cluster does inhibit BMSC progenitor cells not to differentiate then how are osteocyte numbers increased? A profile of all the involved genes during BMSC differentiation from WT, -23aC and decoy mice could go along way in helping to understand the phenotype.

Fig 4e without evidence of protein-protein interaction data (showing that miR-23aC disrupts Prdm16 interaction) the model is somewhat speculative.

Minor concerns:

Figures 1a: Y axis will be FPKM instead of RPKM.

Response to the reviewers' comments:

We appreciate the reviewers finding our study interesting and providing constructive suggestion to complete our study of novel mechanism of miR-23a cluster in regulating Prdm16/TGF- β signaling in osteocyte differentiation. In this revision, we performed further analyses and experiments to answer the questions raised by reviewers. Major changes were highlighted in yellow in the manuscript and all "Extended Data Figures" have been changed to "Supplementary Figures".

Reviewers' comments:

Reviewer #1 (Remarks to the Author):

This study finds that gain-of-function (GOF) and loss-of-function miR-23a transgenes have opposing effects on osteocyte density. The authors then attempt to link these changes to alterations in Prdm16, which is a negative regulator of TGFbeta signaling. While many of the results are interesting, they are in general over-interpreted. Additional studies are required to support the conclusions drawn by the authors, especially the models proposed in figure 4.

1. The tissue-specificity of each transgene that has an effect on bone mass needs to be verified. Along the same lines, if multiple independent lines were generated for each transgenes, the authors need to indicate if the bone phenotype was present in each line.

Response

The transgene Col1a1 2.3kb promoter has been used to generate several osteoblast specific transgenic mice. In the first paper reporting this transgene promoter¹, the authors generated Col1a1-LacZ reporter mice (2300lacZ) and found the expression pattern shown by X-gal staining was restricted to osteoblasts in bones and odontoblasts in incisors (Figure 2 in the paper¹). They also generated Col1a1-luciferase reporter mice (2300lucif) to detect expression level in adult (Table II in the paper¹). The highest expressed tissues are bones, teeth and the tail. Minor expressions of luciferase were detected in skin and tendon. The expression in the other tissues was hardly detectable. Our lab used the same promoter to generate these transgenic mice², and found similar expression pattern in osteoblasts (Figure 3 in the paper²). To further verify the specificity of the expression in our Col1a1-miR-23aC transgenic mice, we collected different tissues including calvarial bones, livers, spleens, lungs, kidneys, hearts, brains and skins from Col1a1-miR-23aC and WT littermates. Most

tissues express WT levels of the individual microRNAs in the miR-23a cluster except skins which is expected to over-express transgenes based on previous reports. See below for the result.

We also generated two independent transgenic mouse lines for each transgene (Supplementary Fig. 6). All these independent lines exhibit similar bone phenotypes by μ CT with the same transgenes. *Col1a1-miR-23aC*, *Col1a1-miR-23D* and *Col1a1-miR-27D* mice showed low bone mass in spines and femurs. *Col1a1-miR-24D* mice do not have significant bone phenotype. (Also see page 4, paragraph 1 in the manuscript).

2. Both GOF and LOF transgenes resulted in low bone mass but had different effects on osteoblast number and osteocyte density. However, no explanations or possible mechanisms responsible for these discordant results were provided or discussed. Along the same lines, the results shown in figure 2i demonstrate that the GOF and LOF transgenes have opposite effects on *Sost* mRNA levels, at least in vitro. If similar levels of regulation occur in vivo, one would have expected opposing effects on osteoblast number or bone

formation when comparing the GOF and LOF mice. However, this was not the case. Some explanation or discussion of this lack of effect is required.

Response

While the number and surface of osteoblasts in GOF mice were decreased significantly (Fig 2a, b), there were no significant differences in LOF mouse lines (Supplementary Fig. 8). Since osteocyte numbers were increased in GOF mice but decreased in LOF mice (Fig. 2g, h), we speculated that the miR-23a cluster promotes osteoblast to osteocyte transition (the model of Fig. 4f) and overexpression of miR-23a cluster in GOF depletes osteoblast pools causing decrease of bone formation.

However, the low bone mass phenotypes in GOF and LOF mouse models resulted from different mechanisms because the number of osteoblasts didn't change significantly in LOF models. We performed histomorphometric analysis for calcein double labeling in LOF lines and found mineral surface (MS)/ bone surface (BS) was significantly decreased in the trabecular bones of the spines in these LOF lines (Supplementary Fig. 9). Mechanistically, TGF- β activity is decreased in LOF mice (Fig. 4b). The disruption of TGF- β signaling has been reported to cause decreased bone formation by several groups^{3,4}. Therefore, the mechanism responsible for low bone mass phenotype in LOF mice is probably due to the functional defects of osteoblasts.

3. The authors suggest that Sost is an important indirect target of the miRNAs mentioned in this study. However, Sost mRNA measurements were exclusively from cultured cells. Sost mRNA levels in bones tissue from the transgenic mice needs to be measured to confirm the in vitro results.

Response

As suggested by reviewer's comment, to check the in vivo expression of Sost, we collected 3-month-old long bones (including femur and tibia) and found that the expression of Sost in Col1a1-miR-23aC mice was significantly elevated, while its expression in Col1a1-miR-23D and Col1a1-miR-27D mice were comparable to WT littermate (Supplementary Fig. 10). This Sost expression results are correlated to the density of osteocytes in the femur cortical bones (Supplementary Fig. 7). We observed more osteocytes in Col1a1-miR-23aC mice but no significant difference in Col1a1-miR-23D and Col1a1-miR-27D mice. Overall, the analysis of in vivo and in vitro expression of Sost (Fig. 2g, h) are consistent. The expression level of

Sost in LOF lines didn't show difference compared to WT littermate perhaps due to compensatory function by other microRNAs.

4. Structural and histological analyses of the skeleton are overly limited. The effects of the transgenes on cortical bone mass should be included as well as the osteocyte density in cortical bone, at least for the GOF mice.

Response

The cortical bone thickness of GOF mice did not show significant difference compared to the WT littermates by μ CT analysis (Supplementary Fig. 3). Further analysis of the osteocyte density in the cortical bones of the GOF mice, we found significantly elevated osteocyte density as observed in trabecular bones (Fig. 2g, h).

5. On page 7, the authors conclude that the changes in *Sost* mRNA contribute to the changes in osteocyte density. However, *Sost* KO mice do not display a change in osteocyte density, at least in cortical bone (JBMR 24:1651, 2009). Therefore, their conclusion is not supported by experimental evidence.

Response

*In our study, we used *Sost* mRNA level as a surrogate marker for osteocyte density since it is one of the predominant genes expressed in osteocytes. Therefore, the conclusion in the previous report (JBMR 24:1651, 2009) does not change our conclusion that miR-23a cluster promotes osteocyte differentiation.*

6. Discussion of the ChIP results shown in figure 3e is misleading. In contrast to what is stated by the authors, over-expression of *Prdm16* does not lead to more occupancy of the *Sost* locus by this factor but instead either no change or perhaps lower levels. Therefore, the conclusion that *Prdm16* is directly responsible for the reduced histone acetylation and reduced *Sost* expression is not supported by these results. Moreover, ChIP results from a control region or control locus should be presented in the Supplementary section.

Response

*We proposed that *Prdm16* and *Smads* are required for binding to the regulatory region of *Sost* and that is supported by increased occupancy with increased TGF- β and *Prdm16* when either are compared to the control group. However, the trend toward decreased occupancy observed in TGF- β only and TGF- β + dox treated group could be due to sequestration⁵ of*

Prdm16 by interaction with other factors. We included this in the manuscript (see page 8, paragraph 1). We also added the 3'UTR of Gapdh as the negative control region of the ChIP assay in Supplementary Fig. 13. As we expected, the occupancy of Prdm16 was found to be very low compared to regulatory region of Sost, which supports the specificity of Prdm16 recruitment.

7. Inhibition of TGF β signaling in vivo increased bone volume and reduced osteocyte density, but did not alter osteoblast number. There is no explanation for how this maneuver increased bone mass. The authors need to address how this antibody affected their proposed target of TGF β signaling, Sost, in vivo. Moreover, if neither bone formation nor Sost levels were affected by the antibody, then their model shown in figure 5e does not accurately reflect their results and should be modified.

Response

We performed histomorphometric analysis on osteoclasts by TRAP staining and assessed bone formation rate by calcein double labeling on the mice treated with anti-TGF- β antibody (1D11) or control antibody (Supplementary Fig. 14). We found the numbers and surface of osteoclasts were significantly decreased in Col1a1-miR-23aC mice with 1D11. Furthermore, there was a trend of increasing mineral apposition rate and mineralizing surface in Col1a1-miR-23aC mice with anti-TGF- β antibody compared to control antibody. These histomorphometric analyses along with Fig. 4d suggest TGF- β signaling affects different aspects of bone homeostasis by combination of increasing bone formation and decreasing bone resorption in Col1a1-miR-23aC.

The elevated level of TGF- β signaling and Sost expression level was observed in other studies such as of Nek8 mutant mice⁶. Consistent with our observation of 1D11 rescue, the increased expression level of Sost mRNA was normalized to wild type level after 1D11 treatment (Figure 11A in the paper⁶). Therefore, with their result and ours, Sost is one of the downstream targets of TGF- β signaling as we show in Fig. 5e.

8. A section in the methods should be included to describe the statistical analysis.

Response

We added a section describing the statistical analysis according to the reviewer's comment. (See the last section in Methods)

9. A minor point is that it is unclear in the methods section what the 5-7 mm refers to when describing the histological analysis. Is it tissue size, section thickness, or something else?

Response

5-7 mm is the section thickness. This is added to the methods section of the manuscript according to the reviewer's comment (See Histomorphometric analysis section in Methods).

Reviewer #2 (Remarks to the Author):

Osteocytes are the terminally differentiated cell type of the osteoblastic lineage. Osteocytes serve important functions in skeletal homeostasis including controlling osteoclastogenesis and biomechanical sensing. Although the transcriptional regulation of osteoblast differentiation has been well-characterized, the factors that regulate differentiation of osteocytes from mature osteoblasts are poorly understood.

Here, the authors show that the microRNA cluster miR-23a~27a~24-2 promotes osteocyte differentiation with osteoblast-specific gain-of-function (Col1a1-miR23aC) mice showing low bone mass associated with decreased osteoblasts but increased osteocyte numbers whilst loss-of-function transgenic mice overexpressing microRNA decoys for either miR-23a or miR-27a, but not miR24-2, showed decreased osteocyte numbers. RNA-sequencing analysis of bones from the gain-of-function Col1a1-miR23aC transgenic mice showed altered transforming growth factor β (TGF- β) signaling involving Prdm16 transcription factor, a negative regulator of the TGF- β pathway was involved in the regulation of osteoblast differentiation. Prdm16 was directly repressed by the miR-23a cluster with concomitant alteration of Sclerostin (Sost) expression in osteocytes. Inhibition of TGF- β activity using neutralizing antibody, 1D11, rescued the low bone mass and osteocyte phenotypes observed in the gain-of-function

Col1a1-miR23aC transgenic mice. Taken together, the miR-23a cluster regulates osteocyte differentiation by modulating the TGF- β signaling pathway through targeting of Prdm16.

This manuscript is well written and clearly structured. Experimental design is excellent with a novel but clear proposed mechanism and appropriate models used to validate findings and statistics appropriately applied. Figures are generally of high quality and presented in a logical fashion and conclusions drawn are appropriate to the interpretation of results in terms of the experiments conducted. The manuscript outlines the role of miR23a cluster in the regulation of terminal differentiation of osteoblasts to osteocytes via a mechanism involving TGF beta and its upstream regulator Prdm16.

There are, however some few points of consideration that could generally help improve the paper.

1. The authors to further outline the central role of TGF beta signaling pathway in the increase in osteocyte numbers in the Col1a1-miR-23aC mice, treated these mice with an anti-TGF- β 1 antibody (1D11) or control antibody (13C4). The low bone mass phenotype was rescued by 1D11 to levels comparable with those seen in WT mice treated with 13C4 (Fig. 4c). Bone histomorphometric analysis showed decreased osteocyte density in Col1a1- miR-23aC mice after 1D11 treatment (Fig. 4d); however, no significant change in osteoblast number was found following 1D11 treatment (Supplementary Fig. 5). Would have been interesting to see if these changes in osteocyte numbers (decrease) and bone mass phenotype were also observed in Col1a1- miR-23aC mice with specific TGF beta knockout in osteoblast.

Response

We appreciate your thoughtful suggestion of using the genetic rescue approach. There are three isoforms of TGF- β ligands (TGF- β 1-3) and two TGF- β receptors (Tgfbr1&2) in mice. All of these global null mice result in embryonic or perinatal lethality⁷. To circumvent disruption of the indispensable roles of these components in TGF- β signaling during development, we would have to use osteoblast-specific Cre line (such as Col1a1-Cre) to generate conditional knockout of these components. We would expect to observe similar phenotype as we have seen in anti-TGF- β antibody (1D11) treatment experiments. However, this is out of our scope of this study and we believe we have shown the importance of TGF- β using multiple complementary approach that does not make a genetic rescue essential for the conclusions here.

2. There is no list of keywords.

We will add keywords in the process of manuscript submission.

Reviewer #3 (Remarks to the Author):

Nature Communication Review

Criteria:

Major claims: novel, of interest, conclusions original, if not provide refs

What additional is needed

Subjective- will it influence the field

Summary:

This manuscript describes the functional activities of the miR-23a-27a-24-2 cluster related to regulation of bone homeostasis by osteocytes. There are numerous papers in the literature (~22) documenting the effects of the cluster in vitro regulating ESCs, several hematopoietic lineages and osteoblast, muscle differentiation. Using a transgenic model expressing the entire cluster under control of the Col1a1 promoter for examining functional effects in osteoprogenitors and developing osteoblasts, the authors observed a phenotype of low bone mass, decreased osteoblasts and increased osteocytes, and was interpreted as the cluster promoting osteocyte differentiation. Inhibition of each of the miRNAs in the cluster was carried by a decoy strategy that also resulted in low bone mass mediated by miRNAs -23a and -27a; however, miR-24-2 which was the highest expressed (shown in Supplementary), had no LOF effect. The manuscript then goes on to characterize a mechanism focusing on the highest

detected pathway, Prdm 16 -TGFB, by transcriptome RNA-Seq analysis between WT and tg GOF mouse. This is not novel as the linkage of this cluster to TGFB signaling has been reported in several studies in different cell types and (see e.g Chhabra review, PMID20815877). Further a Prdm-1 - TGFb1 axis, has also been previously described. Nonetheless much knowledge can be gained from in vivo studies and particularly for the first time in bone. The authors have generated two mouse models, over-expressing the miR-cluster and inhibiting each miRNA by a decoy method, but these are not fully characterized and indicate complex of systemic effects on bone mass, due likely to potential non-specific /off target effects. Although the Col1a1 2.3 is reasonably bone specific, any cell population expressing high levels of the cluster can potentially secrete the miRNA cluster and can effect non-osseous cells. Thus the findings are not readily interpretable.

Will this study be of general interest and influence the bone field - is questionable as the conclusion of the manuscript that inhibition of osteoblasts, yet more osteocytes. in the differentiation of osteocytes, is difficult to comprehend with fundamental biology that osteocytes form from surface osteoblasts. If TGFB is contributing to the phenotype additional evidence that for either the osteoblast to osteocyte conversion, or the direct phenotypic changes that occur in the osteocytes form the. While many experiments have been performed, and although they can explain, in part loss of bone, overall this study is not sufficiently rigorous to have a significant impact due to problems with the mouse models, data inconsistencies and interpretations which would be misleading without additional studies and detailed explanations of alternative interpretation.

One of the problems with presentation of the study is not having the benefit of adequate discussion of the data because of the format/style of Nature COMM publications.

Major concerns:

1. Related to non-specific effects in the tg mouse. The average FPKM for the miR-23a cluster is around 33 fragment per kilobase per millions. In Supplementary Fig. 1d showing a range of 2.5 to 4.5-fold overexpression for the cluster members in mature osteoblasts. Therefore, the overall overexpression will be very broad and repress all possible, both highly and poorly conserved targets (approximately 300 genes differentially expressed in your study).

Response

Based on the prediction of Targetscan Mouse database, there are 752, 907, 413 potential targets (with highly or poorly conserved sites) of miR-23a, miR-27a and miR-24-2, respectively. Among 300 differentially expressed genes in the GOF mouse model, only 12, 15, 3 genes are significantly downregulated with predicted targets of miR-23a, miR-27a and miR-24-2, respectively (Supplementary Dataset 3). This indicates only few targets are regulated by the miR-23a cluster in mature osteoblasts. The expression of the rest genes might be regulated by indirect interaction with miR-23a cluster. All these genes may work synergistically to lead to the bone mass and osteocyte phenotypes. To identify the most dominant targets contributing to the phenotype, we used IPA pathway analysis and chose Prdm16 for further study. However, we do not exclude the possibility that other genes could also affect osteocyte differentiation.

As for overexpression of microRNAs, 2.5-4.5 folds of increased expression is not really dramatically increase when compared to in vitro overexpression studies but in vivo, this may be sufficient for specific effects as is seen in the case of transcription factor haploinsufficiency diseases. On the contrary, in our previous study using the same Col1a1 transgenic promoter⁸, the expression level of miR-34c was achieved to over 100 fold (Fig. S2A in the paper⁸). An osteocyte phenotype was not observed in that transgenic mouse model supporting specificity, and the osteoblast phenotype in that model wasn't observed until 6 months of age. In addition, another independent study using the same Col1a1 transgenic promoter to overexpress miR-206⁹ only observed mineralization defects but not the phenotype in osteoblast numbers or in osteocyte density in vivo (Fig. 4 in the paper⁹). Therefore, overexpression of the miR-23a cluster has specific osteocyte and osteoblast

phenotypes when compared with over expression miR studies using this promoter in other reports.

2. There are numerous gaps in data where comparisons cannot be made and inconsistencies that are not explained in the paper. Examples: The extended Figure 2 shows in GOF in females at 3-month very with very little change in BV/TV trabecular bone in femur when compared to main Fig 1c spine data. Specific LOF of miR-23a and miR 27a in extended Fig 2 does not show BV/TV for comparisons to spine in Fig 1e.

Response

Axial (spines) and appendicular (femurs) bones differ from the shapes, the mechanical force they bear, the molecules involved in development and the rate of remodeling¹⁰. It is very common to see inconsistencies between spines and femurs in various situations. Furthermore, sex also makes difference¹¹. As for the age-related bone loss, female mice showed greater change in trabecular bone than male mice did (Table1 in the paper¹¹). Therefore, in our GOF mouse model, the greater change in spines is reasonable and suggests that miR-23a cluster plays a more important role in spines.

We thank the reviewer for pointing this out, thereby allowing us to clarify. In previous Extended Data Figure 2c-e, we showed the parameters of trabecular bones of spines in LOF mice (the sample groups in Fig. 1e). We didn't include μ CT data of femurs in LOF mice because we would like to focus on the study of spines as most of the histomorphometric analysis were performed with spine samples. To prevent the readers being confused, we moved Extended Data Figure 2c-e to Supplementary Fig. 4a-c.

3. The absence of any analyses of cortical bone in the femur is a major deficiency in the study given that this is the major osteocyte tissue. The periosteal surface would be very informative related to osteocyte "differentiation" that can be analyzed in absence of the bone marrow environment

Response

In response to the reviewer's comment, we included more analyses of the cortical bone in the femurs.

First, we provided μ CT data of the cortical thickness in femurs of GOF and LOF in Supplementary Fig. 3. However, no significant difference was found in any group. (Also see the last sentence in page 3)

Second, we quantified the osteocyte density with femur cortical bones of the Col1a1-miR-23aC, Col1a1-miR-23D and Col1a1-miR-27D mice in Supplementary Fig. 7. We found the osteocyte density increased in Col1a1-miR-23aC mice but not significantly changed in Col1a1-miR-23D and Col1a1-miR-27D mice. This suggested that miR-23a cluster regulates osteocyte differentiation in both trabecular and cortical regions while LOF of either miR-23a or miR-27a is sufficient to affect osteocyte differentiation in trabecular bones of spines but not enough in cortical bones of femurs. This may be due in part to the rate of turnover and bone formation in different bones and the effect sizes of the different genetic manipulations. (Also see page 5, paragraph 1 in the manuscript).

Third, the calcein double labeling for the dynamic bone histomorphometry (Supplementary Fig. 11) didn't show any significant difference of MAR on the endosteal surface in any GOF and LOF group. Since the cortical thickness was not altered significantly in all groups (Supplementary Fig. 3), we think that it is very likely the mineralization of periosteal is also not changed in all groups.

4. Fig 2c micrograph requires far more detail to explain the interpreted phenotype. More evidence is needed to determine if there is a conversion from OBs to OTs which is higher in miR-23aC model. At higher magnification can pre osteocyte be seen with some extension beginning the transition into a lacunar osteocyte in mineralized matrix. It would be far more informative to show calcein labeling in long bone with rates on periosteal and endosteal surfaces. The irregularity of the calcein labeling on trabecular surfaces is due to osteoclast activity.

Biologically osteocytes must originate from osteoblasts, unless there is some magical way more osteocytes (Fig. 2h).

Response

We added more explanation about Fig 2c according to the reviewer's comment (See page 4, paragraph 2 in the manuscript).

The mineral apposition rate of endosteal surface of long bones were added to Supplementary Fig. 11. However, the rates on periosteal surface are not available but expected to be unchanged due to the insignificant difference of cortical thickness in all groups.

The irregularity of the calcein labeling may result from osteoclast activity. But the activity of osteoclasts is not significantly different in GOF mice. Irregularity of the calcein labeling is not really increased in GOF mice.

The reviewer makes a good point that osteocytes must come from osteoblasts. Therefore, accelerated differentiation of osteocyte would deplete the osteoblast pool and this is the reason why we see decreased osteoblast numbers in GOF mice.

5. Inhibitors of mineralization. True to osteocyte, that inhibitors on mineral deposition are increased in membranous bone of PN d3 mice (Fig 2i; but in figure 2j BMSC from GOF mice show a bit less mineralization. But the more revealing biological information is that osteoblast differentiation is NOT inhibited in BMSCs, if there is significant mineralization detected suggesting that the striking loss of bone in vivo is due to systemic effects and not cell autonomous effects due to miR cluster expression. More clarity in the phenotype would be very much appreciated by analysis of the early markers of the osteoblast, e.g collagen, alkphos, or some early transcription factors known to promote differentiation than is presented in extended Fig. 4a for osteocalcin only.

Response

Because there are not enough strong data to support inhibition of osteoblast differentiation in GOF mice, we decided to remove our hypothesis that miR-23a cluster suppresses osteoblast differentiation and we focus on the effects of miR-23a cluster on the osteoblast-osteocyte transition.

6. The decoy mice - more information is needed. Images of their bone (uCT) should be shown. Evidence that the decoy is working is needed. This reviewer does not have expertise with the procedure and only Refs 10, 11 are provided. Is the decoy targeting the cytosolic

precursor or the mature miRNA? No confirmatory evidence is shown that the miR-decoys actually bind and inhibit the expression of miR-23a or 27a.

Response

The uCT images of spines from each decoy mouse line were added in Supplementary Fig. 4 according to reviewer's comment.

Decoys are synthetic RNA molecules with multiple microRNA binding sites and act as competitive inhibitors for endogenous microRNAs¹². They sequester mature miRNAs from their target mRNA (Figure 1 in the paper¹²). Recent study¹³ also showed decoy transgenic mice can be generated and expressed in osteoblastic cells with Col1a1 3.6 promoter to suppress function of miR-433 (Figure 8 in the paper¹³). In Fig. 3c, it has shown that overexpressing miR-27a decoy can relieved the suppression of miR-27a on the luminescence of the luciferase reporter with a miR-27a binding site (Prdm16 3'UTR) in HEK293T cells. In addition, in Fig. 3d, induced expression of miR-27a decoy increased luminescence of the same reporter with miR-27a binding site in MC3T3-E1 cells. These data suggest decoy can be utilized to inhibit microRNA functions.

7. Fig 3. Bioinformatics and Prdm-TGFB axis. Panel a: How robust is this analysis? Are the data representative of multiple experiments or from long bones? Was this data compared to RNA seq from miR-23a or -27a decoy mice? A rationale that Prdm16 is the one among all the targets potentially carrying over the miR-23a cluster regulation in osteocytes should be provided. Was Prdm16 the only one that was detected by RNA-Seq? Supplementary data showing of the top 10-20 most highly differentially expressed (up and down regulated) and the top 5 most enriched pathways would be desirable information.

Response

IPA analysis is a set of algorithm for generating hypothesis from a large data set such as RNA-Seq result. We didn't generated RNA-Seq data from decoy mouse lines and used only samples from GOF mice to identify key components leading to the phenotype. There are several potential upstream regulators identified by IPA analysis, but among them, Prdm16 is the only potential target of miR-23a cluster and hence a potential direct as opposed to secondary affect. Prdm16 is also known to be a negative regulator of TGF- β , which is the highest affected pathway in RNA-Seq profile. Therefore, we choose Prdm16/ TGF- β to target in this study.

The TGF- β reporter mice with GOF of the miR-23a cluster showed higher luminescence signals (Fig. 4a, b) suggesting TGF- β signaling is upregulated by miR-23a cluster. The in vitro differentiation of BMSC from GOF mice (Fig. 3f) supported that Prdm16 affects osteocyte phenotype in GOF mice. The ChIP assay in MC3TC-E1 cells (Fig. 4f) connected TGF- β signaling with Prdm16. Given these positive downstream data, we believe the IPA provided a robust analysis of the data set.

According to the reviewer's comments, we added the top 10-20 most highly differentially expressed (up and down regulated) in Supplementary Table 1 and the top 5 most affected pathways in Supplementary Table 2 (see page 6, paragraph 2 in the manuscript).

8. Figure 3. Cell studies are performed in 3 different cell lines. Panel c should have used MC3T3 osteoblasts for the reporter assay and panel d use BMSC (non-differentiated osteoblasts) which should be compared to MC3T3 osteoblasts that were used for the ChIP studies. Alternatively the PN3 calvarial bone (representing largely an osteocytes) from WT are TG mice are endogenous models of decreased and increased Prdm, respectively, not requiring transfection, and the phenotype could be rescued by TGFB antibody. Performing Q-PCR for the levels of TGFB, Prdm16, miRNA levels and osteocyte markers would contribute to the understanding functional relationships.

Response

We performed reporter assay in MC3T3 cells and showed similar interaction of miR-27a and Prdm16 3'UTR (Fig. 3d and see page 7, paragraph 1 in the manuscript).

There are two purposes of the experiment in Fig. 3d. The first is to assess whether overexpression of miR-23a cluster can accelerate osteocyte differentiation. Sost expression is used as a readout of osteocyte differentiation and, indeed, the expression level of Sost is significantly higher in 23aC group compared to WT group with GFP control virus. The second purpose is to verify whether Prdm16 is one of the key regulators for osteocyte differentiation. In the 23aC group infected with Prdm16 virus, the expression of Sost is significantly decreased compared to the 23aC group infected with GFP control virus and is comparable to the level of WT group with GFP control virus. This suggested that the osteocyte phenotype of GOF mice we observed in vivo is mediated by downregulation of

Prdm16. Therefore, differentiation of BMSC is necessary in this study to clarify the role of miR-23a cluster and Prdm16 in osteocyte differentiation.

In the ChIP assay of Fig. 3f, undifferentiated BMSC or calvarial osteoblasts from TG mice may introduce additional co-factors that would make the results difficult to interpret. To verify direct binding of the Prdm16 on the promoter region of Sost, manipulating stable cell lines with defined molecules was a better option for experiments.

Taken together, we have shown the interaction of miR-23a cluster-Prdm16 with Sost expression in Fig. 3e and the interaction of TGF- β -Prdm16 with Sost expression in Figure 3f with Supplementary Fig. 12c&d.

9. Panel 3c begs the question -What happens to the Prdm16 UTR LUC activity if the reporter is transfected into WT and 27a decoy cells? As a control, responsiveness with mutant Prdm16 UTR would be appropriate.

Response

We have added this experiment in Fig. 3d according to the reviewer's comment. The luminescent signals are significantly increased in MC3T3-E1 cell expressing 27a decoy induced by doxycycline. The mutant Prdm16 3'URT has been shown to lose the responsiveness to miR-27a in Figure 4b of previous report¹⁴.

10. Panel 3d is a crucial experiment to establish a direct link between miR-23a cluster and Prdm16. The experiment lacks several controls. Instead of Prdm16 overexpression, Prdm16 shRNA will be more appropriate. Also to prove miR-23aC direct regulation, instead of overexpressing Prdm16 cDNA, it is necessary to overexpress Prdm16 cDNA {plus minus} 3'UTR. Since miRNAs regulate translation first, then mRNA, a western blot for Prdm16 could be shown as evidence direct regulation.

Response

In both in vivo (Supplementary Table 2) and in vitro experiments (Supplementary Fig.12b), we have shown that upregulation of miR-23a cluster leads to downregulation of Prdm16 in osteogenic lineages. The luciferase reporter assay suggesting a direct link between the miR-

23a cluster (specifically miR-27a) and Prdm16 was repeated by us (Fig. 3c and d) and others (Figure 4b in the paper¹⁴). The translational inhibition of Prdm16 by miR-27a was also shown by the western blot in Figure 4k of the report¹⁴.

11. Fig 4 TGFB reporter mice contribute to identifying a mechanism that involves the miR cluster, bone loss due to TGFB inhibition and increased SOST that inhibits osteoprogenitor cells. If the miR cluster does inhibit BMSC progenitor cells not to differentiate then how are osteocyte numbers increased? A profile of all the involved genes during BMSC differentiation from WT, -23aC and decoy mice could go along way in helping to understand the phenotype.

Response

We agree that we do not have sufficient evidence to support our hypothesis that miR-23a cluster suppresses early differentiation of osteoblast progenitors. We have removed this from our hypothesis and conclusions. The current data suggest that decrease of osteoblast may have resulted from depletion of osteoblast pool due to the acceleration of osteocyte differentiation as one important mechanism. We agree with the reviewer on the suggestion for future study of gene profiles involved with osteoblast differentiation at different time points to address the role of miR-23a in early differentiation.

12. Fig 4e without evidence of protein-protein interaction data (showing that miR-23aC disrupts Prdm16 interaction) the model is somewhat speculative.

Response

The protein-protein interaction of Prdm16 and Smad3 has been previously shown by Warner et al.¹⁵ (Figure 3 in the paper¹⁵) and Takahata et al.¹⁶ (Figure 4 in the paper¹⁶) to regulate TGF- β signaling negatively. The microRNA miR-23a cluster does not disrupt the interaction of Prdm16 and Smad3 but downregulates the expression of Prdm16. Therefore, the availability of the negative regulators is decreased and enhance the opportunity for the signal transducer, i.e., Smad complex, to interact with other positive co-factors.

Minor concerns:

Figures 1a: Y axis will be FPKM instead of RPKM.

We modified this unit according to the reviewer's comment.

Reference for revision

1. Rossert, J. et al. Separate cis-acting DNA elements of the mouse pro-alpha 1(I) collagen promoter direct expression of reporter genes to different type I collagen-producing cells in transgenic mice. *129(5):1421-32.(1995)*
2. Zhou, G. et al. Dominance of SOX9 function over RUNX2 during skeletogenesis. *Proc Natl Acad Sci U S A. 103(50):19004-9. (2006)*
3. Tan, X et al. Smad4 is required for maintaining normal murine postnatal bone homeostasis. *J Cell Sci. 1;120(Pt 13):2162-70 (2007)*
4. Matsunobu, T et al. Critical roles of the TGF-beta type I receptor ALK5 in perichondrial formation and function, cartilage integrity, and osteoblast differentiation during growth plate development. *Dev Biol. 15;332(2):325-38 (2009)*
5. Schmidt, SF et al. Cofactor squelching: Artifact or fact? *Bioessays. 38(7):618-26 (2016)*
6. Liu, S. et al. Role of TGF- β in a mouse model of high turnover renal osteodystrophy. *J Bone Miner Res. 29(5):1141-57 (2014)*
7. Goumans, MJ and Mummery, C. Functional analysis of the TGFbeta receptor/Smad pathway through gene ablation in mice. *Int J Dev Biol. 44(3):253-65. (2000)*
8. Bae, Y. et al. miRNA-34c regulates Notch signaling during bone development. *Hum. Mol. Genet. 21, 2991–3000 (2012).*
9. Inose, H. et al. A microRNA regulatory mechanism of osteoblast differentiation. *106(49):20794-9. (2009)*
10. Berendsen AD and Olsen BR. Bone development. *Bone 80:14-8. (2015)*
11. Glatt, V et al. Age-related changes in trabecular architecture differ in female and male C57BL/6J mice. *J Bone Miner Res. 22(8):1197-207 (2007)*
12. Medina, PP and Slack, FJ. Inhibiting microRNA function in vivo. *Nat Methods. 6(1):37-8. (2009)*
13. Smith, SS et al. microRNA-433 Dampens Glucocorticoid Receptor Signaling, Impacting Circadian Rhythm and Osteoblastic Gene Expression. *J Biol Chem. M116.737890. [Epub ahead of print] (2016)*
14. Sun, L. and Trajkovski, M. MiR-27 orchestrates the transcriptional regulation of brown adipogenesis. *Metabolism 63(2):272-82 (2014).*
15. Warner, DR. et al. PRDM16/MEL1: a novel Smad binding protein expressed in murine embryonic orofacial tissue. *Biochim Biophys Acta. 1773(6):814-20 (2007).*
16. Takahata, M. et al. SKI and MEL1 cooperate to inhibit transforming growth factor-beta signal in gastric cancer cells. *J Biol Chem. 284(5):3334-44 (2009).*

REVIEWERS' COMMENTS:

Reviewer #3 (Remarks to the Author):

Summary of Results: Low bone mass in spine (not cortical bone) of tg expression miR23aClustre. This is rescued TGFB antibody. Prdm16 is a target of TFGB. When Prdm16, a negative regulator of TGFB signaling is repressed by the miR23cluster, TGFB increases and promotes differentiation of osteoblasts to osteocytes. When the cluster is O/E there is acceleration in osteocyte differentiation such that surface osteoblast become depleted leading to a low bone mass. This occurs more effectively in trabecular bone that has high bone turnover, in contrast to cortical bone. The increase in osteocyte density is far more apparent in cortical bone, supporting the phenotype of the MmiR23cluster tg mice.

Response to author and editor re-revised manuscript re-review:

The authors have performed a considerable number of new experiments and provided "missing" information in the manuscript that the laboratory had, but was not reported, such as reporting now that two independent transgenic lines were made for each miRNA transgene, now Supplement in Figure 6. Reviewers 1 and 3 had related comments. The most important comments were addressed, and for reasonable explanations were given. While many of the pursued mechanisms (Prdm1, TGFB signaling) to account for the striking loss of bone phenotype were based on evidences from other systems, the manuscript represents a high impact study given that they have demonstrated in vivo multiple functional activities of the miR-23 cluster in regulating bone homeostasis. The studies represent a paradigm for the coordination of miRNA activities regulating tissue homeostasis in general.

In summary the authors:

1. Provided clearer explanation for low bone mass phenotypes in both GOF and LOF mouse models resulting from different mechanisms (reviewer 1 and 3). New studies (sup Fig 8 and Supplement Fig 9, and emphasizing the contributions of TGFB were added.
2. In addition, confusing and speculative components of the original manuscript were reorganized or deleted, such as on osteoblast differentiation.
3. Cortical vs Trabecular bone: Sost expression measured in cortical bone (Supplement 10) supports histologic density of osteocytes (Supplement Fig 7). However, Sost expression data for spine trabecular bone is not presented in Fig 2 for trabecular (spine) and is needed because the images in Fig 2g, only showing a small area of trabecular bone are not as convincing as the cortical bone images of osteocyte density in Supp. Fig 7.
4. Clarified cortical bone changes: Additional data (Supp3) shows that uCT measurements are not significantly different between WT and miR23C and in Supplement Fig 11 there is no change in bone mineral formation rate in miR23aC.
5. TGFB signaling; Authors demonstrated decreased number of osteoclasts by histomorphometry of TRAP stained cells and a trend to increasing bone formation (calcein labelling MAR). When miR 23aC mice are treated with anti-TGFB antibody. This was concomitant with a modest increase in BV/TV (Fig 4c) but a decrease in osteocyte number (Fig 4 d) indicating a rescue.
6. The interactions between the miR23a cluster and TGFB with Prdm6 are made clearer with additional data (Fig 3d). However, several indicated studies with different controls were not carried out and instead references to literature was provided where the validations had been done.

Minor, but important modifications to be made:

1. Line 209 Fig 4a,b should be Fig 4 b,c
2. Supplement Fig 7 and 10 cortical data should be packaged together and shown with the main figures.
3. Supplement Fig 11 which also shows cortical bone data (mineral apposition rate by calcein labelling) has a serious typographical error in the title – "was significantly changed" should be "was not significantly changed".
4. Histologic images should be shown for data Fig 4 d, as osteocyte density is the one parameter that defines the rescue. Alternatively you can show the decrease in osteocyte density in the

cortical bone of the Id11 treated mice.

5. Add in Fig 4 legend that robust TGFB reporter activity in the head is from active bone formation in the post-natal growing cranium.

Reviewer #3 (Remarks to the Author):

Summary of Results: Low bone mass in spine (not cortical bone) of tg expression miR23aClustre. This is rescued TGFB antibody. Prdm16 is a target of TFGFB. When Prdm16, a negative regulator of TGFB signaling is repressed by the miR23cluster, TGFB increases and promotes differentiation of osteoblasts to osteocytes. When the cluster is O/E there is acceleration in osteocyte differentiation such that surface osteoblast become depleted leading to a low bone mass. This occurs more effectively in trabecular bone that has high bone turnover, in contrast to cortical bone. The increase in osteocyte density is far more apparent in cortical bone, supporting the phenotype of the MmiR23cluster tg mice.

Response to author and editor re-revised manuscript re-review:

The authors have performed a considerable number of new experiments and provided “missing” information in the manuscript that the laboratory had, but was not reported, such as reporting now that two independent transgenic lines were made for each miRNA transgene, now Supplement in Figure 6. Reviewers 1 and 3 had related comments. The most important comments were addressed, and for reasonable explanations were given. While many of the pursued mechanisms (Prdm1, TGFB signaling) to account for the striking loss of bone phenotype were based on evidences from other systems, the manuscript represents a high impact study given that they have demonstrated in vivo multiple functional activities of the miR-23 cluster in regulating bone homeostasis. The studies represent a paradigm for the coordination of miRNA activities regulating tissue homeostasis in general.

In summary the authors:

1. Provided clearer explanation for low bone mass phenotypes in both GOF and LOF mouse models resulting from different mechanisms (reviewer 1 and 3). New studies (sup Fig 8 and Supplement Fig 9, and emphasizing the contributions of TGFB were added.

Response

We appreciate this reviewer's comment.

2. In addition, confusing and speculative components of the original manuscript were reorganized or deleted, such as on osteoblast differentiation.

Response

We appreciate this reviewer's comment.

3. Cortical vs Trabecular bone: Sost expression measured in cortical bone (Supplement 10) supports histologic density of osteocytes (Supplement Fig 7). However, Sost expression data for spine trabecular bone is not presented in Fig 2 for trabecular (spine) and is needed

because the images in Fig 2g, only showing a small area of trabecular bone are not as convincing as the cortical bone images of osteocyte density in Supp. Fig 7.

Response

Isolating RNAs specifically from the trabecular regions of spines to evaluate expression of genes is much more challenging than doing it from the cortical regions of long bones. Although the images in Fig 2g are only small areas of trabecular bones, the quantification results cover the whole regions of spine sections with N=7 in each group. The difference of osteocyte density in trabecular bones is significant and can be seen in Fig. 4d and 4e as well. Therefore, our data should be sufficient to show the significant change of the osteocyte density in spine trabecular bones.

4. Clarified cortical bone changes: Additional data (Supp3) shows that uCT measurements are not significantly different between WT and miR23C and in Supplement Fig 11 there is no change in bone mineral formation rate in miR23aC.

Response

This may be due in part to the rate of turnover and bone formation in different bones and the effect sizes of the different genetic manipulations. Although we saw significant changes in trabecular bones, regulation of miR-23a cluster might not be sufficient to affect bone formation in cortical bones. (Also see line 147 in the manuscript.)

5. TGFB signaling; Authors demonstrated decreased number of osteoclasts by histomorphometry of TRAP stained cells and a trend to increasing bone formation (calcein labelling MAR). When miR 23aC mice are treated with anti-TGFB antibody. This was concomitant with a modest increase in BV/TV (Fig 4c) but a decrease in osteocyte number (Fig 4 d) indicating a rescue.

Response

We appreciate this reviewer's comment. This suggests TGF- β signaling plays an important role in our GOF mouse model.

6. The interactions between the miR23a cluster and TGFB with Prdm6 are made clearer with additional data (Fig 3d). However, several indicated studies with different controls were not carried out and instead references to literature was provided where the validations had been done.

Response

We appreciate this reviewer's comment. In this manuscript, we have performed several in vitro cell assays and in vivo genetic mouse model with the reporter line to link interactions among miR-23a cluster, Prdm16 and TGF- β signaling in osteocyte differentiation. It is also shown in different systems by other laboratories (Ref 14-17). Definitely, more studies need to be done to understand the details of the underlying mechanism.

Minor, but important modifications to be made:

1. Line 209 Fig 4a,b should be Fig 4 b,c

Response

Both GOF and LOF are presented in Fig 4a,b. Therefore, this shouldn't be changed to Fig 4 b,c.

2. Supplement Fig 7 and 10 cortical data should be packaged together and shown with the main figures.

Response

We have moved Supplement Fig 7 (images and quantifications of the osteocyte density) to Fig 2i and 2j. Since we don't have Sost expression in trabecular bones for comparison and we want to emphasize the phenotype in trabecular bones, it would be better to leave Supplement Figure 10 (the Supplement Figure 11 in the final version) in the Supplement Information.

3. Supplement Fig 11 which also shows cortical bone data (mineral apposition rate by calcein labelling) has a serious typographical error in the title – “was significantly changed” should be “was not significantly changed”.

Response

This is corrected according to the reviewer's comment.

4. Histologic images should be shown for data Fig 4 d, as osteocyte density is the one parameter that defines the rescue. Alternatively you can show the decrease in osteocyte density in the cortical bone of the Id11 treated mice.

Response

Histologic images are added in Fig 4d according to the reviewer's comment.

5. Add in Fig 4 legend that robust TGFB reporter activity in the head is from active bone formation in the post-natal growing cranium.

Response

This description is added in Fig 4a legend according to the reviewer's comment.